# Test-Time Graph Search for Goal-Conditioned Reinforcement Learning

**Evgenii Opryshko** [* 1 2] **Junwei Quan** [* 1] **Claas Voelcker** [3] **Yilun Du** [4] **Igor Gilitschenski** [1 2]

## Abstract

Offline goal-conditioned reinforcement learning (GCRL) often struggles with long-horizon tasks, where errors in value estimation accumulate and produce unreliable policies. It is typically assumed that effective long-term planning is infeasible without specialized training. In contrast, our work demonstrates that existing GCRL policies can complete long-horizon tasks when combined with a lightweight, training-free planning wrapper. We find that standard goal-conditioned value functions encode locally consistent geometric structure sufficient for planning. Our approach, Test-Time Graph Search (TTGS), constructs a graph over the offline dataset and employs an adaptive subgoal selection strategy. To address unreliable value estimates during shortest-path search, we propose a novel mechanism that softly penalizes long-distance transitions. Our method incurs negligible computational overhead and requires no additional supervision or parameter updates. On the OGBench benchmark, TTGS significantly boosts success rates across multiple base learners and tasks, with primary gains on challenging long-horizon locomotion tasks where some success rates are improved from near-zero to over 90%, often matching or outperforming methods that require complex auxiliary training. Code and videos can be found at https://ktolnos.github.io/ttgs.

## 1. Introduction

Goal-conditioned reinforcement learning trains agents to reach user-specified goals and offers a framework for extracting diverse behaviors from large datasets. The goal specifi-

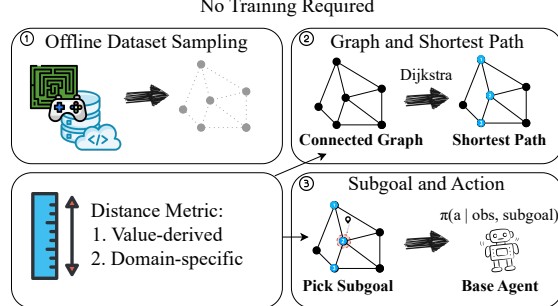

*Figure 1.* **Overview of TTGS.** From an offline dataset, we sample observations to form graph vertices. We assign edge weights using a distance signal, either derived from a pretrained goal-conditioned value function or from domain-specific knowledge. A shortest-path search with Dijkstra's algorithm yields a sequence of subgoals that guides a frozen policy at test time.

cation decouples behavior from specific task rewards, which enables reuse across many objectives and facilitates broad data curation without fragile reward engineering (Schaul et al., 2015; Andrychowicz et al., 2017). In the offline regime, GCRL can leverage precollected data in domains where online interaction is expensive, time-consuming, or unsafe, such as robotics, healthcare, and autonomous driving (Fu et al., 2020; Abdellatif et al., 2021; Kumar et al., 2020).

Despite these advantages, long-horizon decision-making remains challenging. Recent evaluations report that methods performing well in moderate-horizon environments struggle in larger spaces, such as complex mazes (Park et al., 2025; Sobal et al., 2025). To bridge this gap, prior solutions have introduced significant complexity: deploying computationally intensive generative planners (Ajay et al., 2023; Luo et al., 2025), training distributional value function ensembles (Eysenbach et al., 2019), utilizing specialized geometric representations (Baek et al., 2025), or relying on additional data beyond the offline dataset (Emmons et al., 2020). These methods typically operate on the implicit assumption that standard goal-conditioned value functions are inherently too unreliable or noisy to support long-horizon planning directly.

In contrast, we demonstrate that this additional complexity is often unnecessary. We show that standard goal-conditioned

---
[*]Equal contribution [1]Department of Computer Science, University of Toronto, Toronto, Canada [2]Vector Institute, Toronto, Canada [3]University of Texas at Austin, Austin, USA [4]Harvard University, Cambridge, USA. Correspondence to: Evgenii Opryshko <eop@cs.toronto.edu>.

*Proceedings of the $43^{rd}$ International Conference on Machine Learning*, Seoul, South Korea. PMLR 306, 2026. Copyright 2026 by the author(s).

value functions, while prone to long-horizon estimation errors, already encode consistent local geometric structure sufficient for long-horizon planning. By processing this signal with a soft-penalty mechanism, we can guide search over dataset states to yield a path where each segment remains within the capabilities of the pretrained frozen policy, without the need for specialized training. We further show that this framework generalizes to domain-specific heuristics, such as Euclidean proximity.

We introduce Test-Time Graph Search (TTGS), a lightweight, training-free planning wrapper that augments frozen GCRL policies. This design targets the common setting where a practitioner has already invested substantial effort in training and tuning a base goal-conditioned agent, but finds that it fails to reliably solve longer-horizon tasks at deployment. TTGS offers a simple way to improve long-horizon performance by reusing the existing policy and its learned value function without changing the training pipeline.

Unlike prior graph-based reinforcement learning methods that focus on training specialized distance functions or pruning data via online interaction (Emmons et al., 2020), TTGS operates entirely at test time. It constructs a graph over the pretraining dataset, assigns edge costs using an existing distance function, and performs a fast shortest-path search to generate a sequence of intermediate subgoals. The adaptive subgoal selection feeds reachable subgoals ahead of the agent to the frozen policy. We also show that this framework is flexible: while our primary focus is unlocking the latent geometry of value functions, TTGS seamlessly accepts domain-specific heuristics (e.g., Euclidean distance) when available. In both cases, planning over the induced geometry significantly improves trajectory stitching and long-horizon performance without modifying the underlying policy, requiring additional training, or online interaction.

TTGS integrates with existing goal-conditioned learners purely as a test-time procedure. It requires only a distance signal and an offline dataset, adds minimal computational overhead, and preserves the behavior learned by the base policy. The framework complements both hierarchical and non-hierarchical offline GCRL methods by supplying explicit long-horizon search at test time. Since many GCRL algorithms already learn goal-conditioned value functions, the value-derived distance is a natural default that keeps the method simple to adopt.

TTGS is most impactful when three conditions hold: (i) the task requires long-horizon stitching, (ii) the base policy is locally reliable, and (iii) the offline dataset covers intermediate states between the start and goal. These conditions are satisfied by long-horizon locomotion in the OGBench benchmark, where TTGS provides substantial improvements, but not by the manipulation tasks we evaluate, where evaluation

goals lie outside the data manifold. When its conditions are not met, TTGS does not degrade performance but defaults to base policy behavior.

We make the following contributions:

- **Re-evaluation of Value Functions for Planning**: We demonstrate that standard offline GCRL value functions, previously considered too unreliable for search, can support robust long-horizon planning when processed with a novel soft-penalty mechanism.

- **Test-Time Graph Search (TTGS)**: We propose a lightweight, training-free framework that augments frozen agents with graph search. Unlike prior graph-based approaches, TTGS is designed to work with existing policies, requiring no additional supervision or online interaction.

- **Performance Improvement With Minimal Overhead**: We show empirically that TTGS improves the performance of state-of-the-art offline GCRL agents (HIQL, QRL, GCIQL, SAW, and OTA) with negligible computational overhead ($< 1$ second planning time; see Appendix I). On OGBench long-horizon locomotion tasks, TTGS achieves performance competitive with more complex methods that require training or generative planning (Park et al., 2025; Luo et al., 2025; Baek et al., 2025). To support deployment on new tasks, we additionally propose a rollout-free diagnostic, the largest hop ratio along the planned path, that predicts in advance whether TTGS will yield gains.

## 2. Goal-Conditioned Reinforcement Learning Preliminaries

Following Park et al. (2025), we consider the offline goal-conditioned reinforcement learning (GCRL) problem, defined over a controlled Markov process $\mathcal{M} = (\mathcal{S}, \mathcal{A}, \mu, p)$, i.e., a Markov Decision Process (MDP) without rewards, together with an unlabeled dataset $\mathcal{D}$ of trajectories. Here, $\mathcal{S}$ is the state space, $\mathcal{A}$ the action space, $\mu \in \Delta(\mathcal{S})$ the initial state distribution, and $p(s' \mid s, a) \colon \mathcal{S} \times \mathcal{A} \to \Delta(\mathcal{S})$ the transition dynamics. We denote by $\Delta(\mathcal{X})$ the space of probability distributions over a set $\mathcal{X}$.

The offline dataset $\mathcal{D} = \{\tau^{(n)}\}_{n=1}^{N}$ consists of trajectories $\tau^{(n)} = (s_0^{(n)}, a_0^{(n)}, s_1^{(n)}, \ldots, s_{T_n}^{(n)})$ collected by some unknown behavior policy.

The goal in GCRL is to learn a goal-conditioned policy $\pi(a \mid s, g) \colon \mathcal{S} \times \mathcal{S} \to \Delta(\mathcal{A})$ that can efficiently reach any goal $g \in \mathcal{S}$ from any starting state $s \in \mathcal{S}$. Formally, the optimal policy maximizes the expected discounted return:

$$\max_{\pi} \ \mathbb{E}_{\tau \sim p(\tau \mid g)} \left[ \sum_{t=0}^{T} \gamma^t \mathbf{1}\{\|s - g\| < \epsilon\} \right],$$

where $T$ is the horizon, $\gamma \in (0, 1)$ the discount factor, $p(\tau \mid g)$ the trajectory distribution induced by $\pi$, and reward is equal to $\mathbf{1}\{\|s - g\| < \epsilon\}$. The norm $\|\cdot\|$ here defines the benchmark's success criterion: it is computed only during evaluation rollouts and is never accessed during training nor by TTGS at inference. Base learners are trained via hindsight relabeling, which samples goals from dataset trajectories and reduces the reward to a trajectory-position check rather than a metric computation, so no state-space norm is required. Appendix B provides the full per-reward-convention derivation and value-to-distance mappings used by each base learner.

Many GCRL algorithms, including HIQL (Park et al., 2023), GCIQL (Kostrikov et al., 2022), and QRL (Wang et al., 2023), learn a value function $V(s, g) \colon \mathcal{S} \times \mathcal{S} \to \mathbb{R}$ representing the expected discounted reward when navigating from state $s$ to goal $g$.

## 3. Method

We visualize the core challenge that GCRL agents face in Figure 2. In long, complex tasks like maze navigation, agents that attempt to reach far-away goals can get stuck or run off course. However, for shorter horizons, they tend to be reliable. From this observation, we derive the simple key idea behind TTGS: rather than asking a policy to solve a long-horizon task in one shot, we decompose it into short, reliable hops. We do so by equipping test-time planning with a distance signal over states, selecting a compact set of states from an offline dataset, and connecting them into a graph whose edges reflect predicted step costs. At test time, we plan in this graph by computing a shortest path between the start and the goal, then feed the policy a small number of intermediate subgoals along this path. As summarized in Figure 2b, the agent queries all points along the shortest path that are within a certain threshold from the current state, and picks the closest to the goal.

While the framework supports any compatible distance, our primary focus is the general and widely applicable case where the distance is derived from a learned goal-conditioned value function. The following sections describe how distances are calibrated (Section 3.1), how the graph is constructed (Section 3.2), and how subgoal-based planning is executed at test time (Section 3.3).

### 3.1. Distance Prediction

The algorithm begins by obtaining a distance predictor from an available (approximate) metric on the state space. In general, this can be any metric; here we focus on deriving approximate metrics from value function estimates. These predictors need not satisfy formal metric properties (e.g. triangle inequality or symmetry), but we find them reliable

for local reachability; we return to this empirically in Section 3.2.

**Value-derived distance (default).** To enable policy-agnostic planning, we map a goal-conditioned value signal $V(s, g)$ to an expected step distance

$$\hat{d}(s, g) = f\big(V(s, g)\big),$$

where $f : \mathbb{R} \to \mathbb{R}_{\geq 0}$ converts values into an estimate of the environment steps required to reach $g$ from $s$. For the following, common reward conventions, $f$ admits a closed form:

- *Sparse terminal reward.* Reward $1$ only at the goal and $0$ otherwise. Then $V^*(s, g) = \gamma^d$, where $d$ is the shortest-step distance, giving

$$\hat{d}(s, g) = \log_\gamma V^*(s, g).$$

- *Per-step penalty.* Reward $-1$ until reaching the goal and $0$ at the goal. Then

$$V^*(s, g) = -\sum_{t=0}^{d-1} \gamma^t = -\frac{1 - \gamma^d}{1 - \gamma}$$

$$\Rightarrow \quad \hat{d}(s, g) = \log_\gamma\big(1 + (1 - \gamma)\, V^*(s, g)\big).$$

HIQL, SAW and GCIQL use this convention in our experiments.

More generally, any value signal that is monotone with respect to reachability suffices. We clip $\hat{d}$ to avoid negative or infinite outputs. Exact mappings for each base agent are provided in Appendix B.

**Domain-specific distances.** When a task supplies a meaningful geometric or kinematic metric, we can use it directly as $\hat{d}$. This requires no changes to the pipeline below; only the source of edge costs differs. Our experiments include such a domain-specific metric to illustrate the flexibility of the framework.

### 3.2. Graph Construction

Using the distance predictor, the agent now proceeds to build the underlying graph of distances from the pretraining dataset. This graph is goal-agnostic, which means it can be reused for different goals an agent might want to achieve in the environment.

**Vertices selection** Computing all pairwise distances among $N$ states in $\mathcal{D}$ has $O(N^2)$ complexity and may require repeated neural network evaluations, which is prohibitive at scale. We instead sample a compact subset

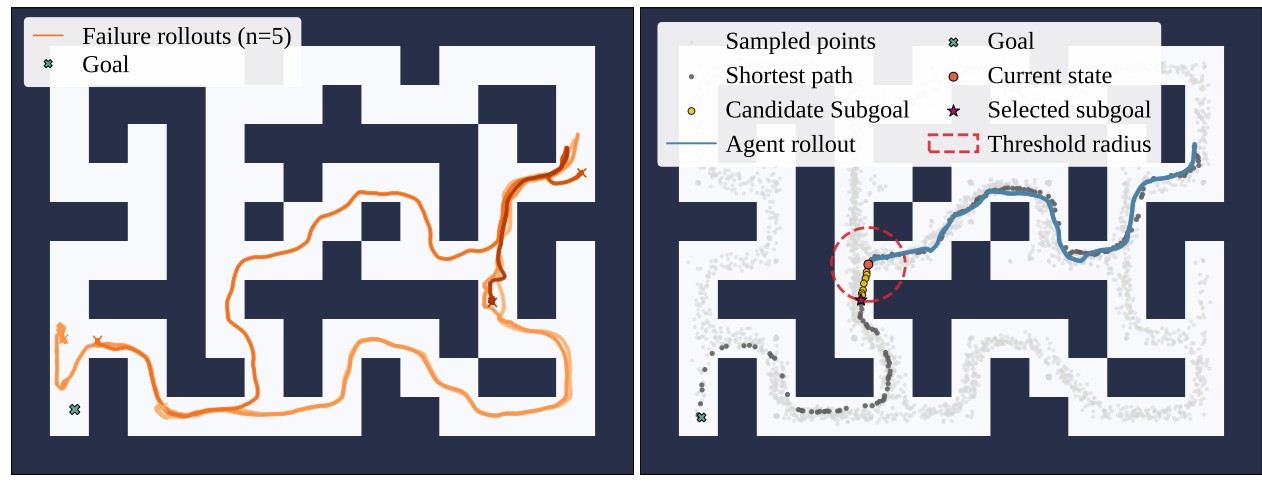

*(a)* Rollouts from HIQL fail to reach a distant goal.       *(b)* TTGS selects reachable subgoals on a guiding path.

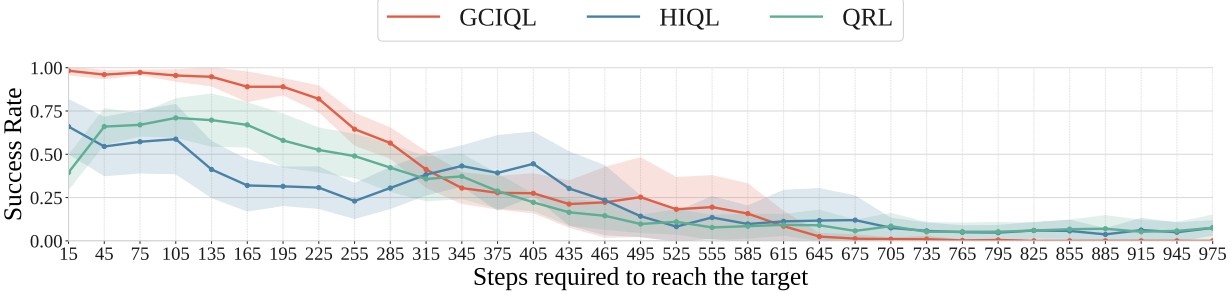

*(c)* Success rate of reaching goals sampled from the expert trajectory and located $n$ steps away from the agent within $1.5n$ step budget in `antmaze-giant-stitch-v0` environment.

*Figure 2.* **Motivation for TTGS:** **(a)** HIQL policy fails to reach a distant goal on `antmaze-giant-stitch-v0`, with multiple attempts failing to exit the starting area and two attempts running out of time due to inefficient path. **(b)** TTGS finds a guiding path using dataset observations. On each step it selects a subgoal which is within a predefined radius from the agent. We mark all data points on the guiding path in gray, and the actual path traversed by the agent in blue. **(c)** Different agents' policy performance decreases as steps required to reach the goal increase. By providing a policy with close subgoals, TTGS improves reliability and efficiency of reaching the goal.

$\mathcal{V} = \{v_i\}_{i=1}^M \subset \mathcal{D}$ and build the graph on $\mathcal{V}$. Uniform random sampling worked well in our experiments. We also tested clustering inspired by Baek et al. (2025) (see Appendix G for details), but it did not produce a statistically significant gain over random sampling, so we adopt random sampling for simplicity.

**Weights derivation** We build a directed, weighted graph $G = (\mathcal{V}, E, \tilde{w})$. Using the distance predictor $\hat{d}$, we compute a matrix $D \in \mathbb{R}^{M \times M}$ with entries $D_{ij} = \hat{d}(v_i, v_j)$ using batched evaluations for efficiency. Because distance predictors can overestimate connectivity across gaps (creating "wormholes"), minimizing raw distance often yields infeasible paths. Conversely, enforcing a hard distance limit risks fragmenting the graph (see Appendix H). To balance these issues, we assign edge weights $\tilde{w}_{ij}$ using a superlinear penalty $p(\cdot)$ for transitions exceeding a trust region $\tau$:

$$
\tilde{w}_{ij} = \begin{cases} D_{ij}, & i \neq j, \ D_{ij} < \tau, \\ p(D_{ij}), & i \neq j, \ D_{ij} \geq \tau, \\ +\infty, & i = j. \end{cases}
$$

Self-loops are removed to avoid unnecessary computations. In experiments we use $p(x) = x \cdot 1000^{x/\tau}$. This penalty preserves local metric structure below $\tau$ while steering the planner toward chains of short, trustworthy hops, utilizing longer edges only when no safer path exists. In practice, triangle-inequality violations of value-derived distances are small within the trust region: among triplets whose pairwise predicted distances all fall under $\tau$, violations average only 0.002–3.1 predicted steps in magnitude across base learners, far below the average violations for all triplets (575–588 steps; see Appendix D).

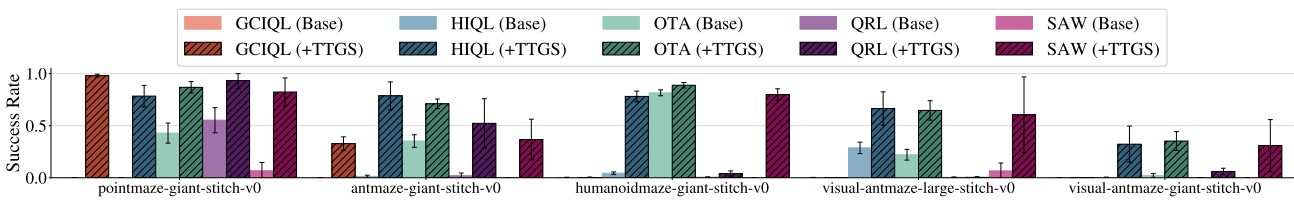

*Figure 3.* **Goal-reaching success rates for QRL, GCIQL, HIQL, SAW and OTA with and without TTGS.** Distances are predicted from each base agent's learned value function. TTGS improves performance on the majority of locomotion tasks that require trajectory stitching. Full results are presented in Table 4.

---

**Algorithm 1** TTGS: Subgoal Selection and Execution

---

**Require:** Graph $G = (\mathcal{V}, E, \tilde{w})$ where $\mathcal{V} = \{v_i\}_{i=1}^M$ are selected states; distance $\hat{d}$; frozen policy $\pi$; start state $s_0$; goal $g$; step budget $T$
1: {Precompute guide path once per episode}
2: $v_s \leftarrow \arg\min_{v_i} \hat{d}(s_0, v_i), \quad v_g \leftarrow \arg\min_{v_i} \hat{d}(v_i, g)$
3: $(p_0, \ldots, p_L) \leftarrow \text{DIJKSTRA}(G, v_s, v_g)$
4: $k_{\text{prev}} \leftarrow 0; \quad s_{\text{cur}} \leftarrow s_0$
5: **while** $g$ not reached **do**
6: $\quad \delta_\ell \leftarrow \hat{d}(s_{\text{cur}}, p_\ell)$ for $\ell = 0, \ldots, L; \quad \delta_g \leftarrow \hat{d}(s_{\text{cur}}, g)$
7: $\quad k \leftarrow \arg\min_\ell \delta_\ell; \quad k \leftarrow \max(k, k_{\text{prev}})$ {index of the closest waypoint ahead}
8: $\quad$ **if** $\delta_g \leq T$ **then**
9: $\quad\quad \tilde{g} \leftarrow g$ {goal is within reach}
10: $\quad$ **else**
11: $\quad\quad \mathcal{C} \leftarrow \{\ell > k : \delta_\ell \leq T\}$ {reachable subgoal candidates}
12: $\quad\quad$ **if** $\mathcal{C} \neq \varnothing$ **then**
13: $\quad\quad\quad \tilde{g} \leftarrow p_{\max \mathcal{C}}$ {take farthest reachable}
14: $\quad\quad$ **else**
15: $\quad\quad\quad \tilde{g} \leftarrow p_{\min\{k+1, L\}}$ {take next subgoal}
16: $\quad\quad$ **end if**
17: $\quad$ **end if**
18: $\quad$ Sample $a \sim \pi(\cdot \mid s_{\text{cur}}, \tilde{g})$ and execute.
19: $\quad s_{\text{cur}} \leftarrow$ next observed state; $\quad k_{\text{prev}} \leftarrow k$
20: **end while**

---

### 3.3. Subgoal Selection and Action Sampling

To find the path to a given goal, the agent proceeds in two steps based on the pre-computed graph.

**Shortest-path precomputation.** At test time, given a start state $s_0$ and a goal $g$, we compute a guide path on $G$ using Dijkstra's algorithm. We first locate the nearest vertices under $\hat{d}$,

$$v_s = \arg\min_{v_i} \hat{d}(s_0, v_i), \qquad v_g = \arg\min_{v_i} \hat{d}(v_i, g),$$

then obtain $(p_0, \ldots, p_L) = \text{DIJKSTRA}(G, v_s, v_g)$. The guide path is computed once per episode and reused throughout execution. Using a fast GPU-based implementation of Dijkstra's algorithm keeps the overhead minimal.

**Adaptive subgoal selection.** During execution, the agent selects subgoals from $(p_0, \ldots, p_L)$ using a step budget $T$. If the current state $s_{\text{cur}}$ is within $T$ steps of $g$ according to $\hat{d}$, the subgoal is set to $g$. Otherwise, the agent identifies the closest waypoint to the current state and updates a progress index $k$ to ensure monotonicity, prohibiting the reference point from moving backward (i.e., $k \leftarrow \max(k, k_{\text{prev}})$). It then chooses the farthest waypoint $p_j$ with index $j \geq k$ that is within budget, $\hat{d}(s_{\text{cur}}, p_j) < T$. If no such waypoint exists, the next waypoint along the path is chosen to ensure forward progress. The frozen goal-conditioned policy $\pi$ is then invoked with current state and selected subgoal. The pseudocode is provided in Algorithm 1.

## 4. Experiments

We evaluate TTGS on OGBench (Park et al., 2025) using three strong offline GCRL baselines provided by the benchmark: QRL, GCIQL, and HIQL (Wang et al., 2023; Park et al., 2023). Additionally, we include SAW (Zhou & Kao, 2025) and OTA (Ahn et al., 2025) to broaden the evaluation. For each dataset, we compare the success rate of the frozen base policy with and without TTGS across five base learners. Every task is evaluated with 50 rollouts, and we report the mean and standard deviation over eight random seeds. In tables, methods within 95% of the best mean are bolded, and cases where TTGS exceeds its corresponding base learner are underlined. All base agents are trained with the public OGBench code using default hyperparameters, and TTGS is applied post hoc without any changes to training.

**Datasets** OGBench provides offline datasets for GCRL, including locomotion domains that test long-horizon and hierarchical reasoning (Park et al., 2025). We use `pointmaze`, `antmaze`, and `humanoidmaze` with `medium`, `large`, and `giant` mazes and three data-collection regimes: `navigate` uses a noisy expert reaching random goals, `stitch` consists of short trajectories that require trajectory stitching, and `explore` contains random exploratory rollouts (available only for `antmaze`). For `antmaze` and `humanoidmaze`, `visual` variants with pixel-based observations are also available.

*Table 1.* **HIQL+TTGS on state-based datasets using value-derived and position-based ($L_2$) distances**. The simple Euclidean distance between body positions, normalized by the average dataset step length, is often sufficient to yield strong gains.

| Dataset | HIQL | HIQL+TTGS-value | HIQL+TTGS-$L_2$ |
|---|---|---|---|
| `pointmaze-giant-navigate-v0` | 43.0 ± 10.5 | **70.9 ± 12.2** | **70.0 ± 9.3** |
| `pointmaze-giant-stitch-v0` | 0.0 ± 0.0 | **80.9 ± 9.0** | **83.0 ± 7.8** |
| `antmaze-giant-navigate-v0` | **65.0 ± 4.1** | 65.8 ± 4.0 | 60.8 ± 4.3 |
| `antmaze-giant-stitch-v0` | 1.4 ± 1.1 | **78.6 ± 13.4** | 60.2 ± 12.0 |
| `antmaze-large-explore-v0` | 2.4 ± 4.4 | **26.6 ± 34.0** | 18.8 ± 23.8 |
| `humanoidmaze-giant-navigate-v0` | 16.0 ± 8.6 | **85.3 ± 6.1** | 74.2 ± 17.5 |
| `humanoidmaze-giant-stitch-v0` | 4.4 ± 1.3 | **78.1 ± 5.1** | 72.6 ± 5.7 |

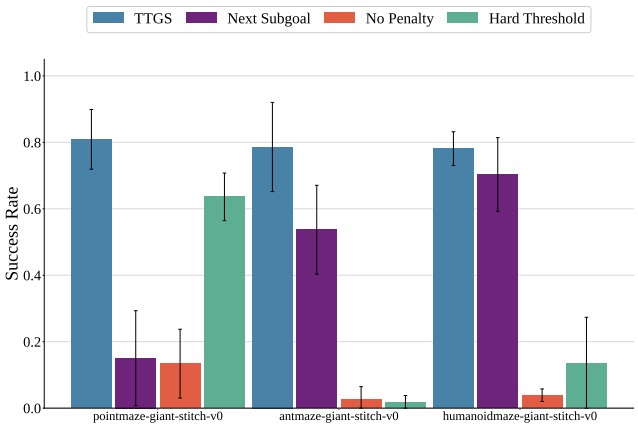

*(a)* Ablations of HIQL+TTGS-value.

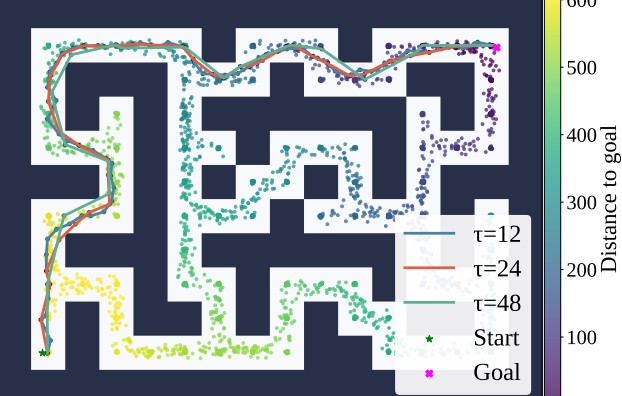

*(b)* Effect of the threshold $\tau$ and predicted distances.

*Figure 4.* **Ablations and hyperparameters.** **(a)** Comparison of full TTGS using HIQL as base learner and value-derived distances with three ablations: *Next Subgoal* replaces our subgoal selection procedure with always picking the immediate next waypoint, *No-Penalty* uses raw predicted distances, and *Hard-Threshold* removes edges with cost $> \tau$. TTGS outperforms both ablations across datasets. **(b)** Effect of penalty threshold $\tau$ on the guide path and a value-derived distance field. Colors denote predicted distances from each dataset observation to the goal in top-right corner. Smaller $\tau$ yields denser subgoals and less direct paths. Larger $\tau$ permits longer hops that can require navigating around obstacles, which is harder for the frozen policy.

## 4.1. Main Results

Figure 3 summarizes results on the largest stitching tasks, which are challenging for existing agents. TTGS uses distances derived from the base agent's value function, so it relies only on information available at test time. TTGS generally improves or maintains success across evaluated environments, with the largest gains on `giant` layouts where one-shot execution is difficult. On `pointmaze-giant-stitch-v0`, TTGS raises HIQL from 0.0% to 80.9% and GCIQL from 0.0% to 98.0%. On `humanoidmaze-giant-stitch-v0`, HIQL is improved from 4.4% to 78.1%. These cases highlight how sequences of short, reliable hops help frozen policies traverse large mazes that require stitching behaviors from disparate parts of the dataset. Notably, TTGS boosts even HIQL, SAW, and OTA which already include a learned high-level controller, suggesting that explicit planning can provide more reliable subgoals than a purely learned subgoal policy in long-horizon settings. We further observe improvements on several pixel-based tasks, indicating that value-derived distances can offer a strong planning signal

even when domain-specific metrics are difficult to define. While Figure 3 highlights the hardest `stitch` tasks, we provide comprehensive results for all evaluated environments, including `medium`, `large`, and `giant` mazes across `stitch`, `navigate`, and `explore` settings, in Table 4 of Appendix A.

To test the generality of TTGS beyond value-based distances, we also evaluate a simple domain-specific distance based on body position. This metric uses the Euclidean distance between agent body positions normalized by the average step length in the dataset. Although it ignores walls, joint configuration and contact dynamics, this signal is sufficient to unlock substantial gains when combined with TTGS. Table 1 reports that HIQL+TTGS with this metric improves performance across most state-based environments, which supports the claim that TTGS can exploit a variety of compatible distances when a value function is unavailable.

*Table 2.* **Success rates on OGBench manipulation tasks (GCIQL).** TTGS matches or improves on the base policy across four tasks; gains are small because the offline data provides few intermediate states between start and goal, and TTGS falls back to the base policy when no reliable plan exists.

| Dataset | GCIQL | GCIQL+TTGS |
|---|---|---|
| scene-play-v0 | 50 ± 7 | 52 ± 4 |
| cube-triple-play-v0 | 4 ± 2 | 4 ± 2 |
| puzzle-4x5-play-v0 | 12 ± 3 | 13 ± 3 |
| puzzle-4x6-play-v0 | 10 ± 0 | 10 ± 0 |

## 4.2. Manipulation

We evaluate TTGS on OGBench manipulation tasks using GCIQL as the base agent, which is the strongest base learner on these domains in the OGBench evaluation. Table 2 reports the resulting success rates. Across the four tasks, gains over the base policy are between 0 and 2 percentage points, and TTGS does not reduce performance on any task.

We attribute this small effect to dataset coverage. In the OG-Bench manipulation datasets, evaluation goals lie outside the main data manifold and the offline trajectories contain few or no intermediate states between start and goal according to the value function. Figure 5 (Appendix C) visualizes this geometry by plotting sampled dataset states on the plane defined by their predicted distances from the start and to the goal: in locomotion datasets the states form a connected band between start and goal, while in the manipulation datasets they cluster far from both. With no intermediate states, TTGS naturally falls back to the base policy.

## 4.3. Predicting when TTGS helps

Motivated by the performance disparity between locomotion and manipulation tasks, we propose the *largest hop ratio* along the TTGS-planned path $(p_0, \ldots, p_L)$ between $s_0$ and $g$ as a *rollout-free* diagnostic for whether TTGS can help on a new task:

$$\rho = \frac{\max(\hat{d}(s_0, p_0), \ldots, \hat{d}(p_{L-1}, p_L), \hat{d}(p_L, g))}{\hat{d}(s_0, g)}.$$

A low $\rho$ means every hop is small relative to the total task (the regime where TTGS helps); a high $\rho$ means at least one hop spans a large fraction of the distance, indicating insufficient intermediate coverage. In our experiments, manipulation hop ratios (0.46–0.62) are 4–6× higher than locomotion ratios (0.04–0.22), and among locomotion tasks the giant mazes (lowest $\rho$) show the largest TTGS gains; the full per-task table is in Appendix F.

The diagnostic is most useful when online evaluation is costly; if rollouts are cheap, simply running TTGS is the faster check. The ratio is also necessary but not sufficient: humanoidmaze-giant-stitch-v0 has $\rho$=0.112 yet

*Table 3.* **TTGS consistently boosts HIQL on long-horizon stitch tasks.** Success rate for HIQL with and without the TTGS on giant-stitch-v0 tasks as we vary $\tau$ and $T$.

| Dataset | $\tau$ | $T$ | HIQL | HIQL+TTGS |
|---|---|---|---|---|
| pointmaze | 12 | 24 | 0.0 ± 0.0 | **78.4 ± 10.3** |
| | 24 | 24 | | 57.2 ± 12.7 |
| | 24 | 48 | | **80.9 ± 9.0** |
| | 24 | 96 | | 62.6 ± 16.7 |
| | 48 | 96 | | 52.0 ± 20.0 |
| antmaze | 12 | 24 | 1.4 ± 1.1 | **78.6 ± 13.4** |
| | 24 | 24 | | 40.9 ± 14.6 |
| | 24 | 48 | | 19.0 ± 9.2 |
| | 24 | 96 | | 2.2 ± 1.6 |
| | 48 | 96 | | 1.1 ± 1.5 |
| humanoidmaze | 12 | 24 | 4.4 ± 1.3 | 52.2 ± 6.1 |
| | 24 | 24 | | 65.5 ± 12.0 |
| | 24 | 48 | | **78.1 ± 5.1** |
| | 24 | 96 | | **79.8 ± 7.7** |
| | 48 | 96 | | **76.8 ± 9.7** |

GCIQL+TTGS shows no improvement because the base policy is unreliable even at short range (cf. Figure 2c).

## 4.4. Ablation Study

We study two design choices: adaptive subgoal selection and the edge-length penalty introduced during graph construction. We also examine sensitivity to the distance penalty threshold $\tau$ and the subgoal selection threshold $T$.

**Subgoal selection.** Our default procedure selects the farthest waypoint on the precomputed route whose predicted distance from the current state is below a threshold $T$, falling back to the next waypoint after the closest node when no such waypoint exists. The alternative always commits to the immediate next waypoint. Figure 4a shows that this simpler strategy degrades performance, especially on pointmaze-giant, while the adaptive rule yields robust improvements. We hypothesize that aiming at the farthest reachable waypoint reduces unnecessary micromanagement of local motion and promotes efficient progress toward reachable, yet sufficiently distant, subgoals.

**Distance penalty.** We compare the dynamic soft penalty from Section 3.2 to a *No-Penalty* variant (raw distances) and a *Hard-Threshold* variant (removing edges > $\tau$). Figure 4a shows that on stitching tasks, the soft penalty significantly outperforms the no-penalty baseline. Furthermore, as detailed in Appendix H, a hard threshold fails on these tasks because noisy value estimates frequently cause graph disconnections. The soft penalty maintains connectivity by assigning high costs to optimistic edges rather than removing them, allowing the planner to recover from local value function errors.

**L2-kNN baseline.** We also compare TTGS to an L2-kNN landmark planner that uses purely Euclidean distance between body positions and Dijkstra search over k-NN graph with the same subgoal-execution rule. L2-kNN reaches a competitive peak (59.4% at $k{=}10$ on `humanoidmaze-giant-stitch-v0`, 66.6% at $k{=}20$ on `antmaze-giant-stitch-v0`) but is fragile to $k$ (low $k$ disconnects the graph, high $k$ admits wormhole-like shortcuts) and the optimum differs across environments. TTGS reaches 76.7% and 76.5% on the same two tasks and $\tau$ is easier to tune than $k$. Full details are in Appendix E.

**Replanning.** We also investigate the effect of online replanning in Appendix J. We find that the initial guide path is generally robust, and frequent replanning typically offers limited benefit.

**Hyperparameters and scaling.** We sweep the edge threshold $\tau$ and the subgoal threshold $T$. Table 3 reports HIQL results on the three largest state-based mazes. TTGS outperforms the base policy or preserves performance across all tested settings, with a moderate penalty threshold and a subgoal selection threshold near $T = 2\tau$ generally working best. Small $\tau$ values bias the planner toward chains of short hops that can lengthen routes, whereas large $\tau$ values admit long edges that the policy may fail to traverse; degenerate settings collapse to a single (start, goal) edge and recover base-learner performance. In practice $\tau$ can be set offline by inspecting the shortest paths produced by TTGS (fragmented paths indicate $\tau$ is too small, paths with long unreliable edges indicate it is too large) and then choosing $T = 2\tau$. We use $M{=}4000$ graph vertices in all main experiments; sweeping $M \in \{125, \ldots, 8000\}$ shows that build time stays under two minutes at $M{=}8000$, shortest-path computation under 1.5 s, and success rate increases monotonically with $M$ (see Appendix I.1).

Finally, Figure 4b visualizes the value-derived distance field together with guide paths produced by TTGS. The distances obtained from the value function reflect the environment's structure, with obstacles forming high-cost barriers and corridors forming low-cost channels. As the edge threshold $\tau$ decreases, the planner trusts only short edges and yields denser routes with more subgoals; as $\tau$ increases, longer edges are permitted, producing sparser but potentially less reliable paths.

## 5. Related Work

Offline goal-conditioned reinforcement learning (GCRL) aims to learn multi-task policies from fixed datasets. A key challenge is to infer reliable goal-conditioned values from imperfect data. Prior approaches stabilize value estimation with expectile regression (Kostrikov et al., 2022; Park et al.,

2023), treat values as quasi-metric distances (Wang et al., 2023), estimate goal-conditioned value functions via contrastive objectives (Eysenbach et al., 2022; Zheng et al., 2024), incorporate temporal abstraction into value learning (Ahn et al., 2025), or impose physics-informed regularization derived from the Eikonal equation (Giammarino et al., 2025). These methods perform well on moderate horizons but often fail on tasks that require trajectory stitching. TTGS is designed to complement them by supplying test-time planning to improve their long-horizon performance.

Graph-based methods for GCRL have been explored along several axes. Some operate online: SoRB (Eysenbach et al., 2019) grows a graph over the replay buffer using a distributional distance function and prunes long edges via a hard distance threshold; GSP (Lo et al., 2024) learns four subgoal-conditioned models for background planning via potential-based reward shaping; and IM-DSG (Bagaria et al., 2025) incrementally builds a skill graph through novelty-driven exploration. SGM (Emmons et al., 2020) builds a sparse graph offline from value-derived edges via two-way consistency, but relies on online edge cleanup. Other methods operate purely offline: VMG (Zhu et al., 2023) abstracts the dataset into a graph-MDP, runs value iteration over learned latent states, and feeds subgoals to a low-level policy; Bagatella & Martius (2023) aggregate goal-conditioned values along graph paths to correct long-horizon noise and use the aggregate as a cost function for model-based MPC over a learned dynamics model; and GSR (Yin & Abbeel, 2024) retrieves sub-trajectories from suboptimal demonstrations via graph search. A related line trains specialized distance learners for planning (Savinov et al., 2018; Zhang et al., 2021; Baek et al., 2025). Relative to these, TTGS operates purely at test time on top of frozen offline GCRL value functions, with no additional training or online interaction; its soft-penalty mechanism preserves connectivity that hard thresholds would destroy (Appendix H), and consistent gains across five base learners show that standard offline GCRL value functions already encode sufficient local geometric structure for planning without specialized distance learning.

TTGS also shares the graph-search structure of motion planning (Kavraki et al., 1996) and TAMP (Garrett et al., 2021), but its edge costs come from learned, noisy value functions rather than known dynamics or collision checkers, which motivates the soft penalty. Other planning paradigms have been explored: diffusion and generative models compose or augment trajectories for long-horizon planning (Janner et al., 2022; Ajay et al., 2023; Luo et al., 2025; Zhang et al., 2025; Lee & Choi, 2025), though they typically demand extra training and computation; model-based planning uses learned dynamics for model-predictive control (Williams et al., 2017; Chua et al., 2018; Hafner et al., 2019; Hansen et al., 2022) or sequence-model planning with trajectory

transformers (Janner et al., 2021). TTGS differs by providing fast ($< 1$s planning time), training-free retrieval of the dataset states, which can complement generative and model-based methods by providing reliable subgoals.

## 6. Limitations

TTGS introduces modest computational overhead: one-time graph construction takes 35–100 s on a single Nvidia L40s GPU with 8 virtual CPU cores, per-episode shortest-path computation typically takes under one second, and per-step action selection adds $\approx 1.5$ ms; see Appendix I for the full breakdown.

**Conditions under which TTGS helps.** As discussed in Section 4.3, TTGS improvements depend on long-horizon stitching being needed, the base policy being locally reliable, and the offline dataset covering intermediate states between start and goal. When these conditions fail, TTGS typically defaults to base-policy behavior without degradation.

**Dataset access.** TTGS requires access at test time to a small subset of the original training data; we use $M{=}4000$, which is $0.4\%$ of an OGBench dataset. When the same user or organization that trained the base policy also deploys it, this data is immediately available. Otherwise, requesting a small sample or collecting one through limited interaction is feasible. We do not require the full offline dataset.

**Failure modes.** Inspecting failed navigation rollouts qualitatively, the dominant failure mode is the base policy failing to execute a short hop near a subgoal (getting stuck or overshooting), not the planner producing infeasible paths. Unlike generative planners that may synthesize physically impossible trajectories, TTGS is conservative by design and relies only on retrieved dataset states.

**Future directions.** When intermediate states are missing, two complementary directions could help: (i) improving value learning so that bridging states already present in the dataset are recognized as reachable rather than high-cost; and (ii) using generative models trained on the offline data to synthesize additional intermediate states. Multiple noisy distance estimates could also be combined for more robust edge weights, alongside better vertex-selection strategies.

## 7. Conclusion

We introduced Test-Time Graph Search (TTGS), a lightweight planning framework that augments pretrained goal-conditioned reinforcement learning agents. TTGS leverages only the offline dataset and a distance signal to construct a graph, then applies efficient search to generate subgoal sequences for a frozen policy. The approach requires no retraining, no additional supervision, and no online interaction.

Experiments on OGBench show that TTGS produces substantial gains on long-horizon locomotion, lifting success rates from near zero to over 90% on the largest stitch mazes, competitive with methods that rely on substantial additional training, generative models, or online interaction. Because TTGS plans only over retrieved dataset states, it is conservative by design: when the offline data lacks intermediate states bridging start and goal, as on the OGBench manipulation tasks, it gracefully defaults to the base policy rather than producing unreliable plans. To support deployment in new domains, we additionally introduce a rollout-free hop-ratio predictor that diagnoses in advance whether TTGS is likely to provide gains on a given task; in our experiments, it cleanly separates the regimes where TTGS does and does not yield improvements.

More broadly, our results suggest that value-based GCRL agents often leave useful geometric structure unused: when the offline dataset covers intermediate states, the value function already encodes local distances sufficient for long-horizon planning. TTGS recovers this structure at test time by reinterpreting values as distances and searching over dataset states, and the same recipe extends to domain-specific signals such as Euclidean body position. We hope this perspective encourages further work on lightweight, training-free planning wrappers for learning-based agents.

## Acknowledgments

Resources used in preparing this research were provided, in part, by the Province of Ontario, the Government of Canada through CIFAR, companies sponsoring the Vector Institute and the Digital Research Alliance of Canada (alliancecan.ca). Any opinions, findings, conclusions, or recommendations expressed in this material are those of the authors and do not necessarily reflect the view of the Canadian government. We thank Seohong Park and co-authors for the OGBench codebase and dataset.

## Impact Statement

This paper presents work whose goal is to advance the field of Machine Learning. There are many potential societal consequences of our work, none which we feel must be specifically highlighted here.

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

# A. Full Results

**Scope.** We present results for TTGS paired with HIQL (Park et al., 2023), GCIQL (Kostrikov et al., 2022), QRL (Wang et al., 2023), SAW (Zhou & Kao, 2025), and OTA (Ahn et al., 2025) in Table 4. We compare against each base learner and two planning methods: GAS (Baek et al., 2025) and CompDiffuser (CD) (Luo et al., 2025). We use value-derived distances for TTGS in this comparison. Results are summarized in Table 4.

**Setting differences.** TTGS operates in a *strictly more constrained* setting than both GAS and CompDiffuser: we require *no additional training, no generative models, and no online interaction* on top of the base learner. We use only the offline pretraining dataset together with the pretrained goal-conditioned value function and frozen policy of each base learner. By contrast, GAS trains a Temporal Distance Representation (TDR) and uses it to train a low-level controller with subgoal supervision, and CompDiffuser trains a diffusion model to synthesize long-horizon trajectories. To the best of our knowledge, TTGS is the first method that operates purely offline and at test time in the offline GCRL setting, so no direct test-time-only baseline currently exists in this category. Despite operating under stricter constraints, TTGS achieves performance broadly comparable to GAS and CompDiffuser on the long-horizon locomotion tasks reported in Table 4.

*Table 4.* **Success rates (%) across datasets.** We report base learners, their TTGS-augmented counterparts, and other planning agents where available. "/" indicates the metric was not reported in the corresponding paper. Means and standard deviations follow the OGBench protocol (50 rollouts per task; averages and s.d. over 8 seeds).

| Dataset | HIQL | HIQL +TTGS | GCIQL | GCIQL +TTGS | QRL | QRL +TTGS | SAW | SAW +TTGS | OTA | OTA +TTGS | GAS | CD |
|---|---|---|---|---|---|---|---|---|---|---|---|---|
| pointmaze-medium-navigate-v0 | 73.6 ± 4.4 | 85.8 ± 5.2 | 51.5 ± 8.2 | 90.9 ± 3.7 | 83.5 ± 3.2 | **98.4 ± 3.4** | 96.8 ± 1.8 | 96.8 ± 1.9 | 80.8 ± 3.1 | 92.4 ± 3.6 | / | / |
| pointmaze-medium-stitch-v0 | 73.0 ± 10.2 | 80.5 ± 20.3 | 18.2 ± 8.9 | 44.0 ± 8.0 | 76.0 ± 8.9 | 93.4 ± 6.8 | 68.2 ± 7.6 | 85.4 ± 6.1 | 60.6 ± 11.2 | 81.0 ± 8.7 | / | **100 ± 0** |
| pointmaze-large-navigate-v0 | 45.2 ± 14.0 | 78.0 ± 9.7 | 32.2 ± 5.8 | **88.1 ± 9.4** | 82.4 ± 5.7 | 92.0 ± 9.7 | 78.0 ± 10.8 | 83.6 ± 6.4 | 85.1 ± 5.7 | 87.7 ± 6.6 | / | / |
| pointmaze-large-stitch-v0 | 13.2 ± 8.0 | 92.2 ± 5.1 | 29.0 ± 5.0 | 29.0 ± 2.2 | 88.8 ± 14.0 | 93.9 ± 8.7 | 41.2 ± 6.7 | 93.2 ± 6.1 | 64.0 ± 14.3 | 69.8 ± 18.5 | / | **100 ± 0** |
| pointmaze-giant-navigate-v0 | 43.0 ± 10.5 | 70.9 ± 12.2 | 0.0 ± 0.0 | **91.9 ± 3.0** | 65.3 ± 10.2 | 88.1 ± 9.5 | 68.5 ± 6.4 | 94.8 ± 2.8 | 67.2 ± 9.8 | 73.6 ± 7.1 | / | / |
| pointmaze-giant-stitch-v0 | 0.0 ± 0.0 | 80.9 ± 9.0 | 0.0 ± 0.0 | **98.0 ± 1.4** | 55.3 ± 12.0 | 93.2 ± 7.7 | 6.8 ± 7.9 | 82.3 ± 13.6 | 42.9 ± 9.6 | 86.8 ± 5.5 | / | 68 ± 3 |
| antmaze-medium-navigate-v0 | **95.2 ± 0.7** | 95.2 ± 1.3 | 72.3 ± 4.7 | 81.1 ± 4.7 | 81.9 ± 10.6 | 83.9 ± 7.8 | 96.3 ± 1.6 | 95.7 ± 1.7 | 95.6 ± 1.5 | 95.6 ± 1.5 | 96.3 ± 1.3 | / |
| antmaze-medium-stitch-v0 | 92.9 ± 2.1 | 95.4 ± 1.3 | 29.5 ± 4.7 | 53.0 ± 8.7 | 62.0 ± 9.4 | 40.1 ± 9.4 | 64.0 ± 5.0 | 94.0 ± 1.9 | 88.5 ± 2.7 | 91.1 ± 3.2 | **98.1 ± 1.2** | 96 ± 2 |
| antmaze-large-navigate-v0 | **90.6 ± 2.5** | 92.3 ± 2.3 | 35.8 ± 2.7 | 57.2 ± 3.8 | 74.0 ± 4.3 | 77.0 ± 3.3 | 88.8 ± 2.5 | 89.6 ± 1.7 | 91.4 ± 1.0 | 91.6 ± 1.4 | 93.2 ± 0.5 | / |
| antmaze-large-stitch-v0 | 73.0 ± 6.0 | 90.8 ± 2.3 | 7.0 ± 2.5 | 30.6 ± 4.6 | 21.0 ± 4.1 | 43.5 ± 10.5 | 3.1 ± 4.8 | 86.2 ± 3.7 | 85.4 ± 3.8 | 91.4 ± 1.5 | **96.3 ± 0.9** | 86 ± 2 |
| antmaze-giant-navigate-v0 | 65.0 ± 4.1 | 65.8 ± 4.0 | 0.4 ± 0.2 | 5.4 ± 2.4 | 11.8 ± 5.2 | 10.8 ± 4.8 | 68.5 ± 3.0 | 71.0 ± 5.3 | 67.4 ± 6.1 | 68.5 ± 4.1 | **76.0 ± 5.9** | / |
| antmaze-giant-stitch-v0 | 1.4 ± 1.1 | 78.6 ± 13.4 | 0.0 ± 0.0 | 32.7 ± 6.6 | 2.0 ± 2.6 | 52.2 ± 23.6 | 0.0 ± 0.0 | 36.8 ± 19.5 | 35.2 ± 6.1 | 71.0 ± 4.6 | **86.2 ± 3.6** | 65 ± 3 |
| antmaze-large-explore-v0 | 2.4 ± 4.4 | 26.6 ± 34.0 | 0.2 ± 0.5 | 66.7 ± 8.3 | 0.0 ± 0.1 | 0.0 ± 0.0 | 1.9 ± 1.8 | 91.8 ± 4.6 | 80.8 ± 5.9 | 80.8 ± 13.9 | 91.0 ± 9.4 | 27 ± 1 |
| humanoidmaze-medium-navigate-v0 | 88.5 ± 3.0 | 93.3 ± 2.9 | 31.8 ± 3.8 | 50.5 ± 5.5 | 19.3 ± 8.1 | 19.7 ± 8.2 | 87.9 ± 2.9 | 86.9 ± 3.1 | 80.8 ± 5.9 | 80.8 ± 13.9 | **96.3 ± 0.4** | / |
| humanoidmaze-medium-stitch-v0 | 86.1 ± 3.0 | 84.6 ± 2.5 | 14.0 ± 3.6 | 20.2 ± 6.8 | 19.4 ± 5.0 | 14.4 ± 3.0 | 63.6 ± 2.2 | 95.0 ± 0.9 | 88.4 ± 3.3 | 83.9 ± 4.9 | **96.2 ± 1.6** | 91 ± 1 |
| humanoidmaze-large-navigate-v0 | 48.0 ± 4.7 | 79.4 ± 5.6 | 2.1 ± 1.1 | 8.4 ± 3.6 | 6.8 ± 2.0 | 10.0 ± 2.7 | 46.8 ± 6.8 | 64.3 ± 6.0 | 78.8 ± 4.3 | 86.4 ± 3.5 | 84.6 ± 3.7 | / |
| humanoidmaze-large-stitch-v0 | 28.6 ± 2.9 | 65.1 ± 10.7 | 0.7 ± 0.5 | 2.0 ± 0.7 | 3.9 ± 2.5 | 3.1 ± 1.5 | 11.6 ± 5.3 | 75.6 ± 12.1 | 58.2 ± 4.9 | 68.9 ± 9.2 | **80.6 ± 2.3** | 72 ± 3 |
| humanoidmaze-giant-navigate-v0 | 16.0 ± 8.6 | 85.3 ± 6.1 | 0.7 ± 0.3 | 0.5 ± 2.0 | 1.1 ± 0.5 | 1.0 ± 1.0 | 40.4 ± 2.7 | 79.0 ± 5.3 | 89.1 ± 3.9 | 92.8 ± 2.1 | 85.7 ± 2.5 | / |
| humanoidmaze-giant-stitch-v0 | 4.4 ± 1.3 | 78.1 ± 5.1 | 0.2 ± 0.3 | 0.2 ± 0.5 | 0.5 ± 0.4 | 4.1 ± 2.4 | 0.0 ± 0.1 | 79.8 ± 5.6 | 81.6 ± 2.8 | 88.8 ± 2.6 | 82.4 ± 2.5 | 67 ± 4 |
| visual-antmaze-large-navigate-v0 | 72.0 ± 3.1 | 83.8 ± 2.6 | 2.4 ± 0.6 | 3.6 ± 0.6 | 0.9 ± 2.4 | 0.3 ± 0.7 | 72.2 ± 4.6 | 75.8 ± 5.2 | 60.2 ± 7.9 | 67.1 ± 12.2 | 85.2 ± 6.6 | / |
| visual-antmaze-giant-navigate-v0 | 5.2 ± 5.2 | 7.6 ± 7.7 | 0.5 ± 0.4 | 0.2 ± 0.3 | 0.1 ± 0.3 | 3.3 ± 3.2 | 4.2 ± 1.2 | 8.4 ± 4.0 | 10.0 ± 6.8 | 19.0 ± 12.1 | 67.2 ± 3.0 | / |
| visual-antmaze-large-stitch-v0 | 28.7 ± 5.5 | 66.4 ± 16.0 | 0.1 ± 0.3 | 0.0 ± 0.0 | 0.4 ± 0.5 | 0.4 ± 0.6 | 6.6 ± 7.6 | 60.4 ± 36.4 | 22.1 ± 5.2 | 64.6 ± 9.3 | 77.2 ± 6.1 | / |
| visual-antmaze-giant-stitch-v0 | 0.2 ± 0.4 | 32.2 ± 17.3 | 0.0 ± 0.0 | 0.0 ± 0.0 | 0.0 ± 0.1 | 6.0 ± 3.1 | 0.0 ± 0.0 | 31.0 ± 24.9 | 2.2 ± 1.9 | 35.2 ± 9.1 | 51.9 ± 6.4 | / |
| visual-antmaze-medium-explore-v0 | 0.2 ± 0.6 | 63.0 ± 28.1 | 0.0 ± 0.0 | 0.0 ± 0.0 | 0.7 ± 1.0 | 0.8 ± 1.5 | 2.4 ± 3.4 | 58.0 ± 26.7 | 30.0 ± 15.5 | 80.5 ± 34.0 | 65.9 ± 6.8 | / |
| visual-antmaze-large-explore-v0 | 0.0 ± 0.0 | 0.8 ± 1.6 | 0.0 ± 0.0 | 0.0 ± 0.0 | 0.0 ± 0.0 | 0.0 ± 0.0 | 0.0 ± 0.0 | 15.2 ± 13.5 | 3.6 ± 3.9 | 25.2 ± 13.7 | 15.1 ± 6.8 | / |
| visual-humanoidmaze-medium-navigate-v0 | 0.8 ± 0.9 | 0.4 ± 0.5 | 0.0 ± 0.0 | 0.0 ± 0.0 | 0.0 ± 0.0 | 0.0 ± 0.0 | 0.1 ± 0.2 | 0.2 ± 0.2 | **1.5 ± 1.4** | 0.9 ± 0.9 | / | / |
| visual-humanoidmaze-medium-stitch-v0 | 0.4 ± 0.5 | 0.7 ± 0.6 | 0.0 ± 0.0 | 0.0 ± 0.0 | 0.0 ± 0.0 | 0.0 ± 0.0 | 0.2 ± 0.3 | 0.2 ± 0.4 | **0.8 ± 0.4** | 0.8 ± 0.7 | / | / |

**Findings.** Despite requiring no additional training, TTGS improves the performance of the base learners in the vast majority of cases. On several hardest tasks (`humanoidmaze-giant-navigate-v0`, `humanoidmaze-giant-stitch-v0`, `visual-antmaze-large-explore-v0`) OTA+TTGS outperforms more complex planning baselines that rely on extra training. These gains support our central claim: simple metric-guided test-time planning can unlock long-horizon competence already latent in value-based GCRL agents.

**Reproduction details and caveats.** The environment provided by OGBench has random start and goal positions even when the task ID and seed are fixed. This explains some variance in the results across tables. CompDiffuser results are taken directly from Luo et al. (2025). GAS results were taken from the official implementation repository https://github.com/qortmdgh4141/GAS, since humanoidmaze results were not reported in the paper.

**Comparison with GAS using fixed hyperparameters.** To compare the sensitivity to hyperparameters with GAS, we compare HIQL+TTGS using a single fixed set of hyperparameters tuned for antmaze across other domains. TTGS uses ($\tau = 24, T = 48$) obtained on antmaze-giant-navigate-v0, GAS uses official implementation with antmaze hyperparameters, both evaluated over 8 random seeds. Results are reported in Table 5. While GAS achieves higher scores on tuned environment, TTGS outperforms GAS on humanoidmaze and pointmaze, highlighting the robustness of TTGS across diverse domains despite using fixed hyperparameters, which might be important in a fully offline setting or when tuning is expensive.

*Table 5.* Success rates on OGBench locomotion tasks for HIQL+TTGS and GAS using a single fixed set of hyperparameters derived from antmaze. While GAS achieves higher scores on antmaze (the domain it is tuned for), TTGS with fixed hyperparameters drastically outperforms GAS on humanoidmaze and pointmaze, highlighting TTGS as a more robust cross-domain planner.

| Dataset | HIQL | HIQL+TTGS (Fixed) | GAS (Fixed) |
|---|---|---|---|
| pointmaze-giant-navigate-v0 | $47 \pm 10$ | $\mathbf{68 \pm 13}$ | $5 \pm 11$ |
| pointmaze-giant-stitch-v0 | $0 \pm 0$ | $\mathbf{73 \pm 14}$ | $0 \pm 0$ |
| antmaze-giant-navigate-v0 | $67 \pm 4$ | $\underline{68 \pm 3}$ | $\mathbf{76 \pm 6}$ |
| antmaze-giant-stitch-v0 | $2 \pm 1$ | $\underline{48 \pm 19}$ | $\mathbf{86 \pm 4}$ |
| antmaze-large-explore-v0 | $5 \pm 7$ | $\underline{60 \pm 26}$ | $\mathbf{91 \pm 9}$ |
| humanoidmaze-giant-navigate-v0 | $14 \pm 6$ | $\mathbf{78 \pm 9}$ | $14 \pm 5$ |
| humanoidmaze-giant-stitch-v0 | $4 \pm 2$ | $\mathbf{69 \pm 18}$ | $8 \pm 4$ |

## B. Base Learners and Value-to-Distance Mappings

**Base learners.** We evaluate five offline GCRL base learners. For HIQL, QRL, and GCIQL we use the OGBench implementations and hyperparameters (Park et al., 2025); for SAW and OTA we use the authors' official code and published hyperparameters. The methods are: **HIQL** (Hierarchical Implicit Q-Learning; Park et al., 2023), a hierarchical extension of IQL that learns a high-level subgoal policy and a low-level goal-reaching policy via expectile regression on goal-conditioned values; **QRL** (Quasi-metric Reinforcement Learning; Wang et al., 2023), which learns a non-negative quasi-metric distance function $d(s, g)$ that approximates the number of steps from $s$ to $g$ and converts it to a policy via a one-step improvement; **GCIQL** (Goal-Conditioned IQL; Kostrikov et al., 2022), a non-hierarchical adaptation of IQL with a single goal-conditioned value head; **SAW** (Flattening Hierarchies with Policy Bootstrapping; Zhou & Kao, 2025), which bootstraps a flat goal-conditioned policy from a hierarchical learner, achieving strong long-horizon performance with a non-hierarchical actor; and **OTA** (Option-aware Temporally Abstracted value; Ahn et al., 2025), which learns goal-conditioned value functions at two temporal resolutions (primitive steps and option transitions) and combines them. We chose HIQL, QRL, and GCIQL because they are the three base learners reported in the OGBench evaluation; we additionally include SAW and OTA as the strong recent offline GCRL methods.

**Subgoal sampling distributions used during base-learner training.** All base learners are trained with their original published hyperparameters and are exposed to long-range goals during training. HIQL and GCIQL mix future-trajectory goals (50%, geometric sampling with mean $\sim$100 steps), current-state goals (20%), and random dataset goals (30%); QRL uses 100% random dataset goals; SAW and OTA use the goal-sampling distributions reported by their authors, which were tuned for long-range goals on this benchmark.

**Hindsight relabeling.** We record the hindsight-relabeling derivation underlying Section 4's claim that the evaluation criterion $\mathbf{1}\{\|s - g\| < \epsilon\}$ is not accessed during training. We illustrate with the per-step-penalty reward convention used by HIQL, SAW, and GCIQL; other conventions (sparse terminal, etc.) admit analogous derivations. For each transition $(s_t, a_t, s_{t+1})$ in $\mathcal{D}$, a goal $g$ is sampled by hindsight relabeling from one of three sources: the current state $s_t$, a future state

along the same trajectory $s_{t+k}$, or a random dataset state. The reward is the trajectory-index identity

$$r_t = \begin{cases} 0 & \text{if } s_t = g, \\ -1 & \text{otherwise,} \end{cases}$$

which involves no norm $\|\cdot\|$ and no state-space metric; the equality $s_t = g$ is an index comparison that holds only when the relabeled goal was sampled to be the current state itself. The TD objective

$$\mathbb{E}\left[\left(V(s_t, g) - r_t - \gamma \mathbf{1}[s_t \neq g] V(s_{t+1}, g)\right)^2\right]$$

has bootstrap target $0$ at the goal, so unrolling the recursion backward yields the fixed point

$$V(s_t, g) \longrightarrow -\sum_{i=0}^{k-1} \gamma^i = -\frac{1 - \gamma^k}{1 - \gamma}$$

for a state $s_t$ that is $k$ steps from $g$ along the offline data, a monotone function of step distance $k$ obtained without computing $\|s - g\|$. Inverting this geometric-sum identity yields the closed-form value-to-distance map used by Section 3.1; per-base-learner maps follow below.

**Value-to-distance transformation for HIQL, SAW, and GCIQL.** HIQL, SAW, and GCIQL use a per-step penalty, that is reward $-1$ until the goal and $0$ at the goal. As discussed in Section 3.1, we map goal-conditioned values to predicted step counts via

$$d(s, g) = \log_\gamma\left(1 + (1 - \gamma) V_{\text{clip}}(s, g)\right),$$

where $\gamma \in (0, 1)$ is the discount factor. To avoid infinite outputs, we clip value predictions to the open interval $\left(-\frac{1}{1-\gamma}, 0\right)$ with a small margin $\varepsilon = 10^{-3}$:

$$V_{\text{clip}}(s, g) = \min\left(-\varepsilon, \ \max\left(V(s, g), -\tfrac{1}{1-\gamma} + \varepsilon\right)\right).$$

We then apply the transform to $V_{\text{clip}}$. For HIQL and SAW we average the two value heads, $V(s, g) = \frac{1}{2}\left(V_1(s, g) + V_2(s, g)\right)$. For GCIQL we use its single value head $V(s, g)$.

**OTA distance.** OTA learns two goal-conditioned value functions at different temporal resolutions: a low-level critic $V^{\text{low}}(s, g)$ defined over primitive environment steps, and a high-level critic $V^{\text{high}}(s, g)$ defined over temporally abstracted option transitions of length $n$ environment steps. Both critics use two value heads; We average them and apply the same clipping to the open interval $\left(-\frac{1}{1-\gamma}, 0\right)$ with margin $\varepsilon$ as above, then inverting the geometric-sum return for the negative reward semantics via

$$\hat{d}(V; \gamma) = \log_\gamma\left(1 + (1 - \gamma) V\right).$$

For the low-level critic this directly yields an environment-step estimate $d_{\text{low}}(s, g) = \hat{d}\left(V^{\text{low}}(s, g); \gamma_{\text{low}}\right)$. For the high-level critic, the value is defined at option boundaries and corresponds to accumulating a penalty $-1$ for each *unsuccessful* option and $0$ for the final successful option; therefore $\hat{d}$ recovers the expected number of failed options $k_{\text{opt}}(s, g) = \hat{d}\left(V^{\text{high}}(s, g); \gamma_{\text{high}}\right)$. This means that $k_{\text{opt}}(s, g) + 1$ is the expected number of options to reach the goal. It can be converted to environment steps by

$$d_{\text{high}}(s, g) = n\left(k_{\text{opt}}(s, g) + 1\right).$$

Finally, we combine both distances into a single environment-step prediction by using $d_{\text{low}}$ for short horizons and gradually switching to the more stable $d_{\text{high}}$ for long horizons, matching the training distributions of both value functions. We use $d_{\text{low}}$ when it predicts distances below $s_0$ (`subgoal_steps`) used to train the actor policies of OTA and linearly interpolate to $d_{\text{high}}$ with $n$ as the ramp width, where $n$ is the `abstraction_factor` hyperparameter of OTA:

$$w = \text{clip}\left(\frac{d_{\text{low}}(s, g) - s_0}{n}, 0, 1\right), \qquad d(s, g) = (1 - w) d_{\text{low}}(s, g) + w \max\left(d_{\text{high}}(s, g), d_{\text{low}}(s, g)\right).$$

**QRL distance.** Quasi-metric RL learns a nonnegative function $d_{\text{qrl}}(s, g)$ that approximates the number of steps from $s$ to $g$. This head already represents a distance, so we use it directly without additional transformation.

**Minimum step constraint.**    In all cases we lower bound predicted distances by 1 step, since no transition can be completed with fewer than one action.

## C. Value Function Geometry

As discussed in Section 4.2, we observe much larger gains from TTGS in locomotion compared to manipulation. We attribute this to the difference in the coverage of the offline dataset. Here we provide the supporting value-function-geometry visualizations.

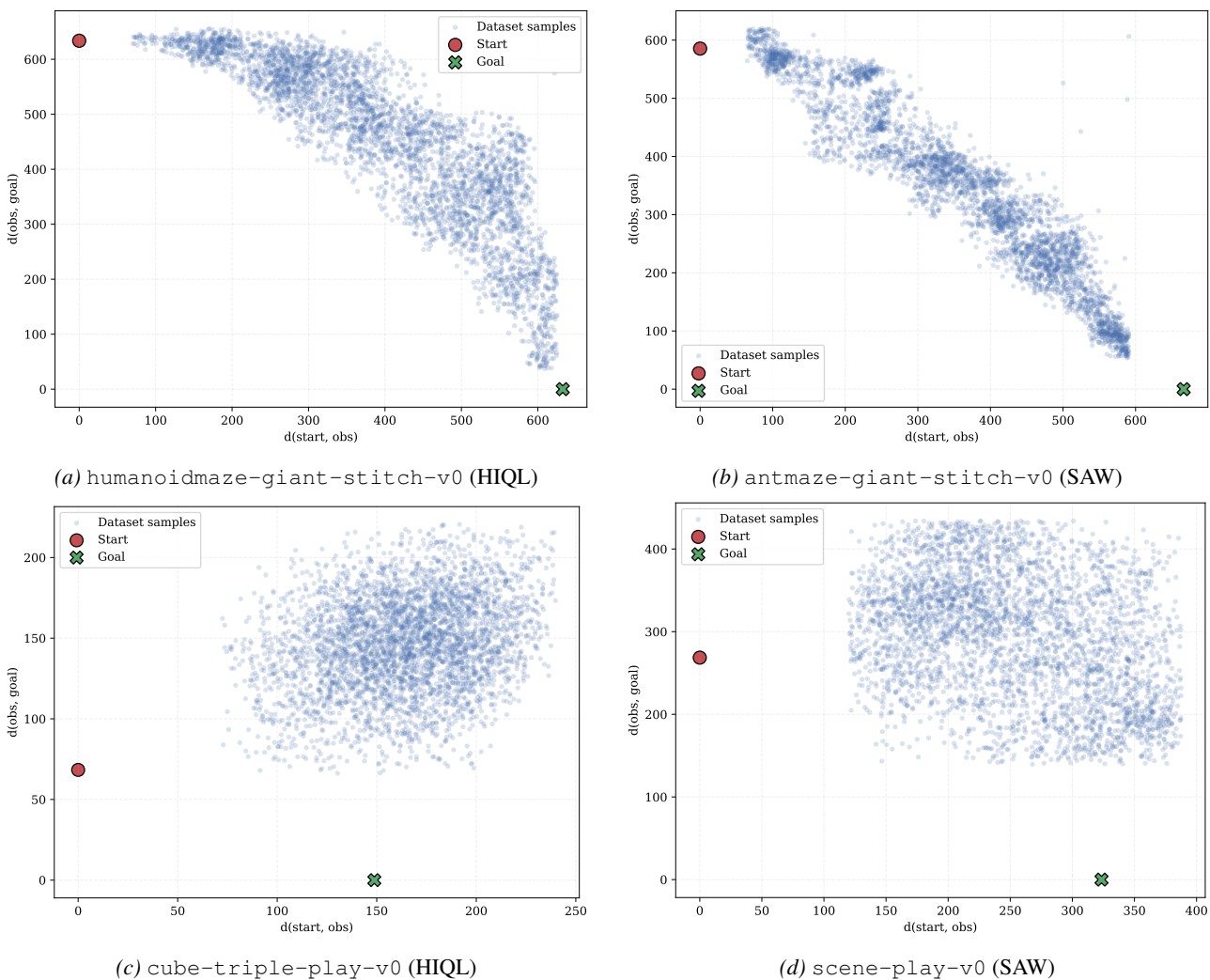

*(a)* `humanoidmaze-giant-stitch-v0` (HIQL)     *(b)* `antmaze-giant-stitch-v0` (SAW)

*(c)* `cube-triple-play-v0` (HIQL)     *(d)* `scene-play-v0` (SAW)

*Figure 5*. **Value Function Geometry.** For each task, we plot sampled dataset states on a 2D plane defined by the predicted distance from the start state ($x$-axis) and the predicted distance to the goal ($y$-axis). In locomotion tasks (a, b), dataset states form a connected band between start and goal (low $x + y$), providing a dense set of intermediate subgoals for graph search. In manipulation tasks (c, d), evaluation goals often lie far from the data distribution, creating a gap where no intermediate states exist. TTGS correctly identifies these high-cost edges and defaults to the base policy rather than hallucinating paths, confirming that it acts as a safe planning wrapper when data support is missing.

## D. Triangle-Inequality Violations of Value-Derived Distances

Value-derived distances are not formal metrics and need not satisfy the triangle inequality (cf. Section 3.1). To quantify how often, and how severely, this property is violated in practice, we sample 1M random triplets globally ("all") and 1M triplets where all pairwise distances fall under the trust region $\tau$ ("under $\tau$"), across three base learners and two giant-stitch datasets,

averaged over 8 seeds. Table 6 reports the violation rate (fraction of triplets with $\hat{d}(a,c) > \hat{d}(a,b) + \hat{d}(b,c)$) and the mean magnitude of the positive violations (in environment steps).

*Table 6.* **Triangle-inequality violations of value-derived distances.** 1M triplets sampled globally ("all") and 1M triplets restricted to pairwise distances under $\tau$ ("under $\tau$"), 8 seeds. Violations within the trust region are small in magnitude (0.002–3.1 steps). For HIQL and GCIQL, this is far below the unconstrained violations (575–588 steps), while QRL exhibits near-zero violation magnitude in both settings. So, non-metricity does not materially distort shortest-path planning within $\tau$.

| | | Violation rate | | Mean positive violation (steps) | |
| --- | --- | --- | --- | --- | --- |
| Base learner | Dataset | all | under $\tau$ | all | under $\tau$ |
| HIQL | `humanoidmaze-giant-stitch` | 0.2% | 13.4% | 585.7 | 3.1 |
| HIQL | `antmaze-giant-stitch` | 1.3% | 9.1% | 587.8 | 1.2 |
| QRL | `humanoidmaze-giant-stitch` | 0.7% | 0.3% | 0.001 | 0.002 |
| QRL | `antmaze-giant-stitch` | 0.7% | 0.2% | 0.001 | 0.002 |
| GCIQL | `humanoidmaze-giant-stitch` | 0.4% | 8.3% | 575.6 | 1.2 |
| GCIQL | `antmaze-giant-stitch` | 5.6% | 8.2% | 574.9 | 1.2 |

Violations do occur within the trust region, but their magnitude is negligible (mean positive violations of 0.002–3.1 steps versus 575–588 for unconstrained triplets). This is consistent with TTGS's strong empirical performance across all base learners despite their substantially different violation profiles: even when the triangle inequality is violated locally, the errors are small enough that they do not meaningfully distort shortest-path planning.

## E. L2-kNN Landmark Baseline: Full Per-$k$ Sweep

In the main text (paragraph "L2-kNN landmark baseline" in Section 4), we summarize a comparison of TTGS to a training-free L2-kNN landmark planner. Table 7 reports the full per-$k$ sweep on the two giant-stitch datasets (HIQL, $M=4000$, 8 seeds).

*Table 7.* **L2-kNN landmark baseline vs. TTGS.** HIQL, $M=4000$, 8 seeds. The disconnect ratio is the fraction of evaluation episodes for which no path exists between the start and goal projections in the kNN graph. The L2 distance is the Euclidean distance between body positions, normalized by the average dataset step length, matching the handling in Table 1. We reuse the TTGS subgoal-selection rule with the same $T$, so the comparison isolates the edge-cost derivation. Optimal $k$ differs across environments ($k=10$ for humanoidmaze, $k=20$ for antmaze); TTGS reaches 76–77% on both with no per-environment $k$ selection.

| Method | $k$ | Success rate | Disconnect ratio |
| --- | --- | --- | --- |
| `humanoidmaze-giant-stitch-v0` | | | |
| HIQL | – | $3.2 \pm 0.5$ | – |
| HIQL + TTGS | – | $\mathbf{76.7 \pm 18.1}$ | 0.00 |
| HIQL + L2-kNN | 2 | $3.2 \pm 1.4$ | 1.00 |
| HIQL + L2-kNN | 5 | $28.1 \pm 20.6$ | 0.64 |
| HIQL + L2-kNN | 10 | $59.4 \pm 15.3$ | 0.00 |
| HIQL + L2-kNN | 20 | $18.8 \pm 13.3$ | 0.00 |
| HIQL + L2-kNN | 50 | $3.2 \pm 3.9$ | 0.00 |
| `antmaze-giant-stitch-v0` | | | |
| HIQL | – | $1.6 \pm 0.9$ | – |
| HIQL + TTGS | – | $\mathbf{76.5 \pm 5.2}$ | 0.00 |
| HIQL + L2-kNN | 2 | $1.6 \pm 1.1$ | 1.00 |
| HIQL + L2-kNN | 5 | $1.6 \pm 1.1$ | 1.00 |
| HIQL + L2-kNN | 10 | $29.2 \pm 23.9$ | 0.60 |
| HIQL + L2-kNN | 20 | $66.6 \pm 3.6$ | 0.00 |
| HIQL + L2-kNN | 50 | $9.6 \pm 7.0$ | 0.00 |

Two observations follow. First, waypoint planning with purely geometric distances does help substantially at the right $k$, confirming that graph-based subgoal decomposition is itself a powerful idea. However, L2-kNN is fragile: at low $k$ the graph disconnects, at high $k$ wormhole-like shortcuts form, and the optimal $k$ differs across environments. TTGS's soft penalty and full-graph construction reach 76% / 77% on both datasets without requiring per-environment $k$ selection. Second, the L2 baseline computes distances between body positions, which presupposes knowing which observation dimensions correspond

to the agent's body, information unavailable in pixel-based domains. Value-derived distances (TTGS's default) require no such privileged knowledge.

## F. Hop-Ratio Predictor: Full Per-Task Table

We define the largest hop ratio $\rho$ in the main text (Section 4.3). The metric is computed by running Dijkstra on the TTGS graph between the nearest-vertex projections of $s_0$ and $g$, then dividing the maximum predicted edge cost along the path (including the entry hop $\hat{d}(s_0, p_0)$ and the exit hop $\hat{d}(p_L, g)$) by $\hat{d}(s_0, g)$. The metric requires only the offline dataset and the value function, with no rollouts. Table 8 reports $\rho$ across all evaluation tasks (GCIQL, 8 seeds), sorted by hop ratio, alongside base and TTGS-augmented success rates.

*Table 8.* **Largest hop ratio $\rho$ across evaluation tasks (GCIQL, 8 seeds), sorted ascending.** Manipulation hop ratios are substantially higher than locomotion ratios (0.46–0.62 vs. 0.04–0.22; about 4x higher on average), confirming insufficient intermediate coverage in the manipulation datasets. Among locomotion tasks, giant mazes have the lowest ratios and the largest TTGS gains. The exception is `humanoidmaze-giant-stitch-v0`, where coverage is good but the humanoid base policy fails at short range, illustrating that low $\rho$ is necessary but not sufficient.

| Dataset | Hop ratio $\rho$ | GCIQL | GCIQL+TTGS | $\Delta$ (pp) |
|---|---|---|---|---|
| `pointmaze-giant-stitch-v0` | 0.041 | 0.0 | 98.0 | +98.0 |
| `antmaze-giant-stitch-v0` | 0.080 | 0.0 | 32.7 | +32.7 |
| `humanoidmaze-giant-stitch-v0` | 0.112 | 0.2 | 0.2 | 0.0 |
| `humanoidmaze-medium-stitch-v0` | 0.140 | 14.0 | 20.2 | +6.2 |
| `antmaze-medium-stitch-v0` | 0.207 | 29.5 | 53.0 | +23.5 |
| `pointmaze-medium-stitch-v0` | 0.217 | 18.2 | 44.0 | +25.8 |
| `scene-play-v0` | 0.458 | 50 | 52 | +2 |
| `cube-triple-play-v0` | 0.470 | 4 | 4 | 0 |
| `puzzle-4x6-play-v0` | 0.620 | 10 | 10 | 0 |

**Recommended procedure for new tasks.** (1) Compute $\rho$ as a rollout-free preliminary check using the offline dataset and the trained value function; thresholds around $\rho \approx 0.4$ separated the regimes where TTGS does and does not yield gains in our experiments. (2) If $\rho$ is low, run TTGS directly: it defaults safely to base-policy behavior when its conditions are not met, so there is no performance risk in trying.

**Necessary but not sufficient.** Even with good coverage, TTGS needs the base policy to execute short hops reliably. `humanoidmaze-giant-stitch-v0` (hop ratio 0.112, GCIQL+TTGS gain 0.0%) illustrates this: the dataset coverage is fine but the humanoid base policy is unreliable even at short range, so TTGS has no reliable executor (cf. Figure 2c). Estimating local competence requires rollouts, at which point it is simpler to evaluate TTGS directly; hence the recommendation in (2) above.

## G. Alternative Training Data Filtering

We tested a vertex sampling scheme based on clustering to improve state coverage, inspired by Baek et al. (2025). While Baek et al. (2025) filter states by temporal efficiency before clustering, we found that adapting this metric to standard value functions was ineffective in preliminary runs. Consequently, we compare standard TTGS (which uses uniform random sampling for graph vertices) against a variant that selects vertices by clustering a large pool of randomly sampled states.

**Clustering.** We sample a large subset $\mathcal{H}$ of 80,000 states uniformly from $\mathcal{D}$. We select graph vertices $\mathcal{V} \subset \mathcal{H}$ using a single-pass greedy clustering rule with radius $r = \tau/2$:

1. Initialize $\mathcal{V} \leftarrow \{s^{(0)}\}$, the first element of $\mathcal{H}$.

2. For each $x \in \mathcal{H}$, compute $m = \min_{v \in \mathcal{V}} \hat{d}(x, v)$.

3. If $m > r$, add $x$ to $\mathcal{V}$. Otherwise assign $x$ to its nearest center and periodically update that center to the member that minimizes total within-cluster distance.

Table 9 compares the performance and runtime of TTGS with random sampling versus clustering. We observe no statistically significant performance gain from clustering, while the computational cost of graph construction increases by an order of magnitude (from ∼40 seconds to >10 minutes). Thus, we adopt random sampling as the default for its efficiency.

*Table 9.* **Effect of state clustering on TTGS.** We compare TTGS with and without state clustering. Clustering uses a random subset of 80,000 states, clustering distance threshold $\frac{1}{2}\tau$, and a maximum of 4,000 cluster centers. From the empirical results, clustering provides only marginal gains while its computational overhead is dominant in the overall cost.

| Environment | HIQL | HIQL+TTGS w/o Clustering | HIQL+TTGS w/ Clustering | Clustering Time (s) | Cluster Centers |
|---|---|---|---|---|---|
| pointmaze-giant-navigate-v0 | 43 ± 12 | **73 ± 12** | **74 ± 9** | 814.76 | 613.0 |
| pointmaze-giant-stitch-v0 | 0 ± 0 | **80 ± 6** | **79 ± 10** | 876.43 | 3904.88 |
| antmaze-giant-navigate-v0 | 64 ± 4 | **67 ± 3** | **69 ± 3** | 706.77 | 4000.0 |
| antmaze-giant-stitch-v0 | 2 ± 1 | **70 ± 14** | **69 ± 11** | 1932.88 | 4000.0 |
| antmaze-large-explore-v0 | 2 ± 4 | **26 ± 34** | **25 ± 33** | 771.11 | 4000.0 |
| humanoidmaze-giant-navigate-v0 | 15 ± 7 | **85 ± 6** | **83 ± 9** | 691.54 | 4000.0 |
| humanoidmaze-giant-stitch-v0 | 4 ± 2 | **78 ± 8** | **78 ± 12** | 705.48 | 4000.0 |

## H. Soft vs. Hard Edge Penalties

We investigate whether the soft penalty scheme is necessary compared to a simpler hard thresholding scheme. We compare our method (Soft Penalty) against a variant where edges with predicted distance greater than $\tau$ are removed entirely (Hard Threshold).

Table 10 presents the results. On simple navigation tasks, the Hard Threshold often performs comparably to the Soft Penalty. However, on tasks requiring trajectory stitching, the Hard Threshold leads to drastic performance drops. For example, success drops from 78.6% to 2.2% on antmaze-giant-stitch-v0 and from 78.1% to 14.0% on humanoidmaze-giant-stitch-v0.

This performance gap is explained by the *Disconnect Ratio*, defined as the fraction of episodes where the goal becomes unreachable from the start state in the constructed graph. Standard value functions are noisy and may underestimate the cost of specific transitions required to bridge disparate trajectories in the dataset. A hard threshold disconnects these essential links, rendering planning impossible. The soft penalty allows these edges to persist with high cost, preserving connectivity while still discouraging their use unless no safer path exists.

*Table 10.* **Comparison of Soft vs. Hard Penalties.** The Hard Threshold variant frequently disconnects the graph on stitching tasks (high Disconnect Ratio), leading to poor success rates. The Soft Penalty maintains connectivity by assigning high costs to optimistic edges rather than removing them. Results are for HIQL+TTGS.

| Dataset | HIQL+TTGS (Soft) | HIQL+TTGS (Hard) | Disconnect Ratio |
|---|---|---|---|
| pointmaze-giant-navigate-v0 | **71 ± 12** | 65 ± 10 | 0.08 |
| pointmaze-giant-stitch-v0 | **81 ± 9** | 66 ± 9 | 0.25 |
| antmaze-giant-navigate-v0 | **66 ± 4** | **67 ± 4** | 0.11 |
| antmaze-giant-stitch-v0 | **79 ± 13** | 2 ± 2 | 0.99 |
| antmaze-large-explore-v0 | 27 ± 34 | **53 ± 32** | 0.00 |
| humanoidmaze-giant-navigate-v0 | **85 ± 6** | 57 ± 9 | 0.47 |
| humanoidmaze-giant-stitch-v0 | **78 ± 5** | 14 ± 14 | 0.84 |
| visual-antmaze-large-navigate-v0 | **84 ± 3** | 64 ± 9 | 0.38 |
| visual-antmaze-large-stitch-v0 | **66 ± 16** | 31 ± 14 | 0.54 |

# I. Runtime Analysis

We provide a detailed breakdown of the computational overhead of TTGS in Table 11. All runtimes were measured on a single Nvidia L40s GPU with 8 virtual CPU cores. Graph construction is performed once per dataset, and the shortest path search is performed once per episode (or sparingly if replanning). Both operations are matrix-heavy and highly parallelizable; we implement them on the GPU. The per-episode planning overhead is generally around 1 second or less. The per-step action selection overhead is negligible ($\approx 1.5$ ms).

*Table 11.* **TTGS Runtime Overhead.** Per-environment cost of graph construction and online planning components in TTGS. Measured on an Nvidia L40s GPU with 8 vCPUs using HIQL as base agent. Graph build is one-time compute for all the tasks, while shortest path is one-time compute per episode.

| Environment | Graph Construction (s) | Shortest Path (s) | Pick Subgoal (ms) | Step Overhead (ms) |
|---|---|---|---|---|
| pointmaze-giant-navigate-v0 | 36.42 | 1.25 | 0.36 | 1.68 |
| pointmaze-giant-stitch-v0 | 34.98 | 0.79 | 0.37 | 1.61 |
| antmaze-giant-navigate-v0 | 36.55 | 0.46 | 0.37 | 1.42 |
| antmaze-giant-stitch-v0 | 36.26 | 0.84 | 0.39 | 1.66 |
| antmaze-large-explore-v0 | 36.40 | 0.46 | 0.38 | 1.33 |
| humanoidmaze-giant-navigate-v0 | 34.99 | 0.59 | 0.38 | 1.39 |
| humanoidmaze-giant-stitch-v0 | 36.35 | 1.0 | 0.39 | 1.38 |
| visual-antmaze-large-navigate-v0 | 100.02 | 0.53 | 0.47 | 2.63 |
| visual-antmaze-large-stitch-v0 | 100.89 | 0.53 | 0.48 | 2.49 |

## I.1. Scaling sweep over the number of graph vertices $M$

To validate that TTGS scales to large offline datasets, we sweep the number of graph vertices $M \in \{125, \ldots, 8000\}$ on the two giant-stitch tasks (HIQL, 8 seeds) and report success rates alongside graph-build and shortest-path times in Table 12. Success rate increases monotonically with $M$. Build time is largely flat for $M \leq 1000$ because the dense $M \times M$ distance matrix fits in a single batched GPU pass and parallelism absorbs the additional work. As $M$ grows beyond what fits in one batch, the matrix must be split across multiple batches and build time grows toward the underlying $O(M^2)$ cost: doubling $M$ from 4000 to 8000 multiplies build time by $\sim 3.5$–$3.8$ (a true quadratic would give 4.0). Even at $M{=}8000$ build time stays under two minutes. Shortest-path time grows roughly linearly with $M$ and stays under 1.5 s.

*Table 12.* **Scaling sweep over $M$.** HIQL, 8 seeds, $M \in \{125, \ldots, 8000\}$ on the two giant-stitch tasks. The dense $M \times M$ distance matrix fits in a single batched GPU pass at small $M$, keeping build time largely flat; for larger $M$ the matrix must be split into multiple batches and build time approaches the underlying $O(M^2)$ cost, staying under two minutes at $M{=}8000$. Shortest-path time stays under 1.5 s; success rate increases monotonically with $M$.

| | humanoidmaze-giant-stitch-v0 | | | antmaze-giant-stitch-v0 | | |
|---|---|---|---|---|---|---|
| $M$ | Success rate | Build (s) | Path (s) | Success rate | Build (s) | Path (s) |
| 125 | 10.4 ± 10.0 | 7.4 ± 2.0 | 0.06 | 29.2 ± 8.3 | 9.3 ± 4.2 | 0.05 |
| 250 | 13.3 ± 11.6 | 5.9 ± 1.1 | 0.07 | 29.9 ± 12.4 | 8.2 ± 4.4 | 0.06 |
| 500 | 42.4 ± 22.9 | 5.0 ± 1.5 | 0.14 | 51.6 ± 12.1 | 3.7 ± 0.4 | 0.10 |
| 1000 | 60.0 ± 19.6 | 6.9 ± 1.7 | 0.24 | 53.6 ± 16.8 | 5.8 ± 1.4 | 0.20 |
| 2000 | 72.8 ± 13.0 | 10.7 ± 1.7 | 0.44 | 70.9 ± 10.6 | 10.2 ± 1.3 | 0.39 |
| 4000 | 76.7 ± 18.1 | 26.2 ± 1.5 | 0.84 | 76.5 ± 5.2 | 25.5 ± 0.9 | 0.69 |
| 8000 | 78.7 ± 12.8 | 99.8 ± 8.7 | 1.31 | 81.1 ± 2.5 | 88.6 ± 2.5 | 1.13 |

# J. Replanning Analysis

We evaluate a variant of TTGS that triggers a full recalculation of the shortest path if the agent deviates from the current subgoal by more than $2T$. To prevent excessive path oscillations, we limit replanning to occur at most once every 50 steps. Table 13 compares this against the standard one-time planning approach. Results show that while replanning can help in

datasets with highly noisy trajectories (e.g., `antmaze-large-explore`), the adaptive subgoal selection on the fixed guide path is sufficient and often superior for most tasks.

*Table 13.* **Effect of replanning in TTGS.** Comparison between the default TTGS (one-time planning) with a variant that replans when the agent strays $> 2T$ from the current subgoal (at most once every 50 steps). *Replan Episode Ratio* indicates the fraction of episodes where replanning occurred. In most environments, the one-time adaptive guide is sufficient, and frequent replanning can degrade performance by causing oscillations.

| Environment | HIQL | HIQL+TTGS w/o Replan | HIQL+TTGS w/ Replan | Replan Episode Ratio |
|---|---|---|---|---|
| `pointmaze-giant-navigate-v0` | $50 \pm 13$ | $\mathbf{73 \pm 12}$ | $\mathbf{72 \pm 6}$ | 0.01 |
| `pointmaze-giant-stitch-v0` | $0 \pm 0$ | $\mathbf{80 \pm 6}$ | $\mathbf{80 \pm 6}$ | 0.00 |
| `antmaze-giant-navigate-v0` | $66 \pm 3$ | $\mathbf{67 \pm 3}$ | $\mathbf{71 \pm 2}$ | 0.52 |
| `antmaze-giant-stitch-v0` | $1 \pm 1$ | $\mathbf{70 \pm 14}$ | $\mathbf{67 \pm 13}$ | 0.38 |
| `antmaze-large-explore-v0` | $2 \pm 3$ | $\underline{26 \pm 34}$ | $\mathbf{50 \pm 32}$ | 0.07 |
| `humanoidmaze-giant-navigate-v0` | $14 \pm 5$ | $\mathbf{85 \pm 6}$ | $78 \pm 4$ | 0.52 |
| `humanoidmaze-giant-stitch-v0` | $4 \pm 3$ | $\mathbf{78 \pm 8}$ | $33 \pm 10$ | 0.98 |
| `visual-antmaze-large-navigate-v0` | $46 \pm 14$ | $\mathbf{82 \pm 7}$ | $70 \pm 10$ | 0.86 |
| `visual-antmaze-large-stitch-v0` | $18 \pm 8$ | $\mathbf{78 \pm 4}$ | $53 \pm 24$ | 0.76 |

## K. Use of Generative AI

LLMs were used to revise and polish writing on a single-paragraph scale.

## L. Hyperparameters

TTGS does not require training and has only three hyperparameters, so tuning is quick and simple. We did not run extensive sweeps; Table 14 lists the values used in our experiments.

$M$ is the number of dataset states sampled as graph vertices. Larger $M$ increases coverage and cost. In our preliminary experiments we found $M = 4000$ to be sufficient while performant, with $M$ as low as 100 helpful for improving base agent's performance.

$\tau$ is the edge-length threshold used to compute graph weights: distances greater than $\tau$ are penalized. A larger $\tau$ permits longer hops and is suitable only when long-range distance estimates are accurate and the pretrained policy can reliably traverse them.

$T$ is the maximum allowed distance between the current state and the selected subgoal at execution time. Choose $T$ near the range where the frozen policy is most reliable.

**Hyperparameter selection methodology.** We selected $(\tau, T)$ using a lightweight manual sweep rather than an extensive grid search. In practice we evaluated a small set of candidate pairs following a doubling schedule, e.g., $(\tau, T) \in \{(12, 24), (24, 48), (48, 96)\}$, which keeps the search space compact while covering substantially different hop lengths. Usually, we tried 1 or 2 hyperparameter settings per dataset and base agent. For each candidate $(\tau, T)$, we first constructed the TTGS graph and qualitatively checked the resulting shortest paths: (i) whether paths contain "blank" segments indicative of poor connectivity or unreliable long edges, and (ii) whether paths become excessively fragmented into too many short hops (suggesting an overly conservative threshold). We then ran 1 random seed with promising parameters and selected the pair that produced stable, interpretable paths and the strongest improvement in task performance. We did not tune SAW across environments and still observe reasonable performance. Since the sweep involves only a handful of settings, we encourage users to repeat this procedure for new environments or base agents, as the optimal trade-off between long hops and reliability can vary.

*Table 14.* Hyperparameter settings used in our experiments.

| Dataset | $M$ | HIQL $(\tau, T)$ | QRL $(\tau, T)$ | GCIQL $(\tau, T)$ | SAW $(\tau, T)$ | OTA $(\tau, T)$ |
|---|---|---|---|---|---|---|
| pointmaze-medium-navigate-v0 | 4000 | (24, 48) | (24, 48) | (24, 48) | (24, 48) | (24, 48) |
| pointmaze-medium-stitch-v0 | 4000 | (24, 48) | (24, 48) | (24, 48) | (24, 48) | (24, 48) |
| pointmaze-large-navigate-v0 | 4000 | (24, 48) | (24, 48) | (24, 48) | (24, 48) | (24, 48) |
| pointmaze-large-stitch-v0 | 4000 | (24, 48) | (24, 48) | (48, 96) | (24, 48) | (24, 48) |
| pointmaze-giant-navigate-v0 | 4000 | (24, 48) | (24, 48) | (12, 24) | (24, 48) | (24, 48) |
| pointmaze-giant-stitch-v0 | 4000 | (24, 48) | (12, 24) | (6, 12) | (24, 48) | (24, 48) |
| antmaze-medium-navigate-v0 | 4000 | (24, 48) | (24, 48) | (24, 48) | (24, 48) | (24, 48) |
| antmaze-medium-stitch-v0 | 4000 | (24, 48) | (24, 48) | (24, 48) | (24, 48) | (24, 48) |
| antmaze-large-navigate-v0 | 4000 | (24, 48) | (24, 48) | (24, 48) | (24, 48) | (24, 48) |
| antmaze-large-stitch-v0 | 4000 | (24, 48) | (24, 48) | (24, 48) | (24, 48) | (24, 48) |
| antmaze-giant-navigate-v0 | 4000 | (24, 48) | (12, 24) | (12, 24) | (24, 48) | (24, 48) |
| antmaze-giant-stitch-v0 | 4000 | (12, 24) | (12, 24) | (12, 24) | (24, 48) | (24, 48) |
| antmaze-large-explore-v0 | 4000 | (24, 48) | (24, 48) | (24, 48) | (24, 48) | (24, 48) |
| humanoidmaze-medium-navigate-v0 | 4000 | (24, 48) | (24, 48) | (24, 48) | (24, 48) | (48, 96) |
| humanoidmaze-medium-stitch-v0 | 4000 | (24, 48) | (24, 48) | (24, 48) | (24, 48) | (48, 96) |
| humanoidmaze-large-navigate-v0 | 4000 | (24, 48) | (24, 48) | (24, 48) | (24, 48) | (48, 96) |
| humanoidmaze-large-stitch-v0 | 4000 | (24, 48) | (24, 48) | (24, 48) | (24, 48) | (48, 96) |
| humanoidmaze-giant-navigate-v0 | 4000 | (36, 72) | (24, 48) | (24, 48) | (24, 48) | (48, 96) |
| humanoidmaze-giant-stitch-v0 | 4000 | (24, 48) | (24, 48) | (12, 24) | (24, 48) | (48, 96) |
| visual-antmaze-large-navigate-v0 | 4000 | (12, 24) | (12, 24) | (12, 24) | (24, 48) | (24, 48) |
| visual-antmaze-giant-navigate-v0 | 4000 | (12, 24) | (12, 24) | (12, 24) | (24, 48) | (24, 48) |
| visual-antmaze-large-stitch-v0 | 4000 | (12, 24) | (12, 24) | (12, 24) | (24, 48) | (24, 48) |
| visual-antmaze-giant-stitch-v0 | 4000 | (12, 24) | (12, 24) | (12, 24) | (24, 48) | (24, 48) |
| visual-antmaze-medium-explore-v0 | 4000 | (12, 24) | (12, 24) | (12, 24) | (24, 48) | (24, 48) |
| visual-antmaze-large-explore-v0 | 4000 | (12, 24) | (12, 24) | (12, 24) | (24, 48) | (24, 48) |
| visual-humanoidmaze-medium-navigate-v0 | 4000 | (12, 24) | (12, 24) | (12, 24) | (24, 48) | (48, 96) |
| visual-humanoidmaze-medium-stitch-v0 | 4000 | (12, 24) | (12, 24) | (12, 24) | (24, 48) | (48, 96) |

