# OpenReview forum: "Test-Time Graph Search for Goal-Conditioned Reinforcement Learning"
_ICML.cc/2026/Conference — ICML 2026 regular_

### Official Review · Reviewer_VfQS · 2026-02-21

**Soundness:** 3
**Presentation:** 3
**Significance:** 2
**Originality:** 2
**Overall Recommendation:** 4
**Confidence:** 3

**Summary:**

TTGS introduces a framework to apply planning to offline goal-conditioned reinforcement learning methods. A graph over a sampled subset of states from an offline dataset are used to create a graph, with edge weights derived from value functions or known heuristics. To mitigate issues due to unreliable long jumps in the state-space, a soft penalty is applied to distance estimates over a threshold. At test time, the start and goal state are mapped to vertices on the graph by distance. A graph planning algorithm like Dijkstra is used to find a path of waypoints from the start to goal. Subgoals/waypoints are then selected from the planned path and used to condition the policy to guide the agent towards them. Empirically, TTGS improves the baseline performance over all tested GCRL methods.

**Compliance With Llm Reviewing Policy:**

Affirmed.

**Final Justification:**

The author's rebuttal addressed my concerns.

**Key Questions For Authors:**

1. For each baseline method (HIQL, QRL, etc.) how are the subgoals sampled during training, and what is the distribution of goal distances? I'd like to know whether TTGS's gains persist when the baseline is retrained with a goal-sampling scheme that emphasizes longer horizon goals
2. How does TTGS compare to a simple training-free waypoint planner such as a kNN landmark graph over the sampled states and using A*/Dijkstra + the same subgoal execution rule? Instead of the value-derived distance and soft penalty, use a purely geometric distance and standard sparce graph. This would clarify whether gains from TTGS are mostly from the graph search and waypoints versus value-derived distances and wormhole handling.
3. Are there quantitative predictors that can be used to predict when TTGS would help or not, like graph connectivity metrics or nearest-neighbor distances?
4. Are there environments in which the baselines are already mostly successful where a poor choice of $\tau$ and $T$ would result in worse performance?

**Limitations:**

yes

**Strengths And Weaknesses:**

**Strengths**
- The overall method and framing was easy to follow
- The gains shown across the OGBench environments are quite substantial, with a low barrier to adoption as further training is not needed
- Ablation helps justify that the additions of subgoal choice and penalty to avoid the wormhole phenomenon matters
- I appreciate the care given to explaining the limitations and failure modes

**Weaknesses**
- Missing a baseline for planning over states/landmarks, so its hard to isolate how much of the gains are from a TTGS-specific design or the fact that any waypoint planning helps (see the questions section below for more details)
- Scalability concerns when it comes to requiring many states for sufficient coverage, as both compute and space will grow quite large
- Its unclear how the baselines were trained with respect to their conditioned subgoals and distance away, and how much uplift TTGS actually provides if these were trained with long-range goals in mind
- The ablation over $\tau$ an $T$ suggest that these choice can have a pretty large impact over the performance, and its not clear how these should be chosen other than broad sweeps over a range of values

**Misc**
- Section 2 in the first sentence you have "i.e," but it should be "i.e.,"
- Section 2 you define the expectation over over trajectories $\tau$ but write a sampling distribution over states $\tau \sim p(s|g)$, then immediately in the following text use $p(\tau | g)$. The notation here is mismatched

---

> ### Author Rebuttal · Authors · 2026-03-31
>
> We thank Reviewer VfQS for the constructive review.
>
> > How does TTGS compare to a simple training-free waypoint planner such as a kNN landmark graph over the sampled states and using A\*/Dijkstra \+ the same subgoal execution rule? Instead of the value-derived distance and soft penalty, use a purely geometric distance and standard sparse graph.
>
> We ran this experiment with HIQL, $M{=}4000$, 8 seeds:
>
> | Method | k | Success Rate | Disconnect Ratio |
> | :---- | :---- | :---- | :---- |
> | **humanoidmaze-giant-stitch** | | | |
> | HIQL | -- | 3.2 ±0.5 | -- |
> | HIQL + TTGS | -- | 76.7 ±18.1 | 0.00 |
> | HIQL + L2-kNN | 2 | 3.2 ±1.4 | 1.00 |
> | HIQL + L2-kNN | 5 | 28.1 ±20.6 | 0.64 |
> | HIQL + L2-kNN | 10 | 59.4 ±15.3 | 0.00 |
> | HIQL + L2-kNN | 20 | 18.8 ±13.3 | 0.00 |
> | HIQL + L2-kNN | 50 | 3.2 ±3.9 | 0.00 |
> | **antmaze-giant-stitch** | | | |
> | HIQL | -- | 1.6 ±0.9 | -- |
> | HIQL + TTGS | -- | 76.5 ±5.2 | 0.00 |
> | HIQL + L2-kNN | 2 | 1.6 ±1.1 | 1.00 |
> | HIQL + L2-kNN | 5 | 1.6 ±1.1 | 1.00 |
> | HIQL + L2-kNN | 10 | 29.2 ±23.9 | 0.60 |
> | HIQL + L2-kNN | 20 | 66.6 ±3.6 | 0.00 |
> | HIQL + L2-kNN | 50 | 9.6 ±7.0 | 0.00 |
>
> This experiment is informative in several ways. First, waypoint planning with geometric distances does help substantially at the right $k$, confirming that graph-based subgoal decomposition is a powerful idea. However, L2-kNN is fragile: performance is sensitive to $k$, with graph disconnections at low $k$ and wormhole-like shortcuts at high $k$. The optimal $k$ also differs across environments ($k{=}10$ for humanoidmaze, $k{=}20$ for antmaze). TTGS addresses exactly this fragility through its soft penalty and full-graph construction, achieving 76.7% and 76.5% without needing to select $k$. Second, the L2 baseline computes distances between body positions, which requires privileged knowledge of which observation dimensions correspond to the agent's position. In visual domains this information is simply absent. Value-derived distances, which TTGS uses by default, require no such privileged knowledge and perform comparably to L2 on state-based tasks (Table 1) while being the only option for pixel-based tasks. We also used our subgoal selection procedure for this experiment, which relies on the distance threshold $T$. This would require tuning both distance-based quantity and number of neighbours, while TTGS has two distance-based hyperparameters that are easy to tune together with $T=2\tau$ heuristic.
>
> > For each baseline method (HIQL, QRL, etc.) how are the subgoals sampled during training, and what is the distribution of goal distances?
>
> All base learners are trained with their original published hyperparameters and are exposed to long-range goals. HIQL and GCIQL mix future-trajectory goals (50%, geometric sampling with mean ${\sim}100$ steps), current-state goals (20%), and random dataset goals (30%). QRL uses 100% random dataset goals. SAW and OTA were tuned by their authors for performance on the benchmark with long-range goals in mind. TTGS provides complementary gains on top of these already-tuned baselines.
>
> > Are there environments in which the baselines are already mostly successful where a poor choice of $\tau$ and $T$ would result in worse performance?
>
> Large $\tau$ and $T$ cause the path to degenerate to (start, goal), recovering base learner performance. Very small values force excessive waypoints, but we did not observe complete degradation in any experiment. The heuristic $T = 2\tau$ works well: $\tau$ approximates the inter-subgoal distance, and $T = 2\tau$ selects the next-to-closest subgoal, enabling smooth transitions. We will add this discussion.
>
> > Are there quantitative predictors that can be used to predict when TTGS would help or not?
>
> We conducted a new manual analysis of 100 failed navigation episodes. The dominant cause of failure is the base policy failing to reach a close subgoal, while the planned path is geometrically valid. This suggests a simple predictor: TTGS helps most when the base policy is reliable at short range but fails at long horizons. For manipulation, we analyze graph structure in Appendix C and show that sparse coverage and out-of-distribution goals limit the ability to construct guiding paths. We will discuss both in the main text. Regarding scalability, we provide M-sweep experiments in our response to Reviewer 1VSa showing how performance scales with graph size and computational overhead remains modest (under 2 minutes even at $M{=}8000$).
>
> > Section 2 typos.
>
> Thank you, we will fix these.
>
> We hope we have addressed all raised concerns with new experiments (which we will include in the revised paper) and planned revisions. If any remain, we would be happy to discuss them further. We kindly ask the reviewer to consider updating their score in light of these responses.

---

> > ### Author Rebuttal · Reviewer_VfQS · 2026-04-03
> >
> > Thank you for the response to the rebuttal, my concerns have been addressed. I have revised by score.

---

### Official Review · Reviewer_1VSa · 2026-03-04

**Soundness:** 3
**Presentation:** 3
**Significance:** 3
**Originality:** 2
**Overall Recommendation:** 5
**Confidence:** 4

**Summary:**

TTGS is a wrapper method for Offline-GCRL that improves long-horizon task performance without requiring policy retraining. The approach constructs a graph from the offline dataset's states and uses the learned value function to assign edge costs. A soft-penalty distance metric is employed to penalize transitions unlikely to be executable by the base policy, and the resulting path is computed via Dijkstra's algorithm. The path yields a sequence of subgoals, which are sequentially provided to the frozen policy. Three key design choices are central to the method: (1) value-function-based edge weights combined with soft-penalty weighting rather than hard feasibility thresholds, (2) adaptive waypoint selection that advances to the next subgoal once proximity constraints are satisfied, and (3) zero-retraining applicability across any offline goal-conditioned RL agent.

The paper evaluates TTGS on OGBench across five distinct base learners (HIQL, QRL, GCIQL, SAW, OTA). Notable performance improvements are observed on long-horizon navigation tasks, including pointmaze-giant-stitch (0.0% to 80.9% for HIQL), humanoidmaze-giant-stitch (4.4% to 78.1%), and similar gains on antmaze variants. Computational overhead remains modest across all experimental conditions.

**Compliance With Llm Reviewing Policy:**

Affirmed.

**Final Justification:**

The paper presents Test-Time Graph Search (TTGS), a training-free planning wrapper for offline GCRL agents. TTGS constructs a graph over dataset states, assigns value-derived distances as edge weights, and applies Dijkstra search with a soft penalty for unreliable long-range edges. The approach is straightforward, compatible with a range of base learners, and yields substantial improvements on long-horizon navigation tasks in OGBench.

**Strengths.** Empirical results on long-horizon stitching are strong and consistent across five base learners (HIQL, QRL, GCIQL, SAW, OTA), with success rates increasing from near-zero to over 80% on several giant maze tasks. This consistency suggests the method offers complementary benefits rather than exploiting properties of a specific algorithm. The soft-penalty formulation is well-motivated, with ablations demonstrating clear advantages over both no-penalty and hard-threshold alternatives. The method does not require retraining, which is practically valuable, and the description is sufficiently detailed for re-implementation. Both technical soundness and clarity of presentation are high.

**Weaknesses.** The main limitation is scope: the method is effective for long-horizon navigation but shows little impact on manipulation tasks, which is narrower than initially framed. Conceptual novelty is moderate, as graph-based planning over dataset states has precedent in prior work (e.g., SoRB). The contribution is in the specific test-time techniques for offline GCRL without specialized training, rather than a new conceptual framework.

**Rebuttal assessment.** The rebuttal addressed my main concerns with substantive evidence. The new M-sweep experiments resolved computational scaling concerns and showed that only a small fraction of the offline dataset (approximately 0.4%) is needed at test time. Failure mode analysis across 100 episodes indicated that base policy execution, not graph coverage, is the dominant failure cause. The L2-kNN comparison, added in response to another reviewer, effectively isolated the impact of TTGS’s design choices from generic waypoint planning, strengthening the case for novelty.

In follow-up, I raised two remaining concerns: the framing overstated the method’s scope, and there was no quantitative predictor for when TTGS is likely to help. The authors’ second response addressed both points. They committed to narrowing the framing in the abstract, introduction, and conclusion, and introduced a rollout-free diagnostic (largest hop ratio) that separates navigation from manipulation tasks in their experiments. The two-factor account based on dataset coverage and local base policy competence also provides a principled explanation for the humanoidmaze counterexample.

**Final position.** The rebuttal addressed all original concerns about scaling, failure modes, framing, and applicability predictors through new experiments or concrete commitments. The paper presents a well-executed, practically useful method with strong empirical results. The authors have demonstrated consistent experimental rigor and responsiveness throughout the discussion. I am raising my score from 4 to 5 and recommend acceptance.

**Key Questions For Authors:**

1. Can you provide results on non-maze OGBench tasks, particularly manipulation environments? The current evidence strongly suggests navigation, even though the paper frames contributions more broadly. Clarifying whether the method generalizes to manipulation would be important.
2. What is the empirical computational cost as the dataset size increases? At what N do the O(N^2) or O(Nk) costs become practically problematic? Are there recommended subsampling strategies?
3. How does TTGS compare to other test-time planning approaches, such as tree search with learned world models or RRT-based planning using the value function as a heuristic?
4. When the method fails, what are the primary failure modes? Does failure stem from inaccurate distance estimation, insufficient state-space coverage in the graph, or the inability of the base policy to execute the proposed subgoals?

**Limitations:**

The paper would benefit from a more explicit discussion of: (a) the method being fundamentally suited to navigation-like problems rather than general goal-conditioned RL, (b) the need for the full offline dataset at test time, (c) empirical scaling with large datasets, and (d) sensitivity to value function calibration. The current limitations section is brief and does not fully address these points.

**Strengths And Weaknesses:**

## Strengths

* The empirical results on long-horizon trajectory stitching are strong. Improvements from near-zero to over 80% success rates indicate a marked shift in agent capability on these tasks. This scale of improvement is uncommon and merits attention.
* Testing across five distinct base learners yields consistent improvements across all algorithms, suggesting that the method offers complementary benefits rather than exploiting architecture-specific properties of any single algorithm.
* The soft-penalty formulation for edge costs is well-motivated. Hard feasibility cutoffs can discard useful information, while soft penalties allow the planner to consider less certain edges when necessary. Ablation studies provide supporting evidence for this choice.
* The zero-retraining requirement is genuinely valuable for deployment scenarios and has practical significance.
* The writing is clear, the problem formulation is direct, and the method description is detailed enough that re-implementation from the paper seems feasible.

## Weaknesses

* The main limitation is the narrow range of tasks in the evaluation. Although OGBench includes manipulation tasks, the strongest results are limited to maze-based navigation problems rather than general offline goal-conditioned RL. This distinction should be made more explicit.
* Graph construction depends on access to the full offline dataset at test time. While this is feasible in research, deployment scenarios rarely include the training dataset with the policy. This practical constraint should be discussed in more detail, as it limits applicability.
* The method’s effectiveness relies on value function calibration. When the value function provides accurate distance estimates, the approach works as intended. However, if the value function is poorly calibrated, especially outside the training distribution, distance estimates can become unreliable, and the planner may select infeasible transitions. Soft penalties help, but do not fully address this issue. Additional analysis of failure modes in value function calibration would strengthen the paper.
* Computational scaling is not empirically validated. While the theoretical complexity is O(N^2) for pairwise distances or O(Nk) for k-NN approaches, practical performance on large offline datasets is not shown. Since modern datasets can be very large, empirical scaling plots relating graph construction time to dataset size would clarify this issue.
* The comparison with GAS and CompDiffuser is asymmetric, as these methods require additional training. Including other test-time-only planning methods or explicitly noting this difference would improve the comparison.
* The framing overstates the method’s scope. The title suggests broad applicability, but the approach requires navigation-style goals and dataset availability at test time. This is a meaningful but narrower domain than the framing implies.
* Ablation studies cover subgoal selection and soft penalties but do not span a sufficiently wide experimental range. Additional ablations could address how sparse the graph can be before performance degrades, whether simpler distance heuristics can replace value-function-based costs, and where TTGS fails and why. More detailed failure-case analysis would strengthen the contribution.
* Graph-based planning over dataset states has prior precedent. Earlier work, such as SoRB (Eysenbach et al., 2019) and related methods, has explored this direction. The main novelty here is in the engineering: soft penalties, adaptive subgoal selection, and broad empirical evaluation. Making an existing idea work well is valuable, but represents incremental rather than conceptual novelty.

---

> ### Author Rebuttal · Authors · 2026-03-31
>
> We thank Reviewer 1VSa for the thorough and constructive review.
>
> > The main limitation is the narrow range of tasks in the evaluation... The framing overstates the method's scope.
>
> Manipulation results are in Appendix C and Figure 5. We will move them to the main text and clarify in the abstract that the primary gains are in long-horizon locomotion. We note that TTGS does not degrade performance on manipulation tasks; it defaults to base policy behavior when the coverage is sparse.
>
> > Graph-based planning over dataset states has prior precedent. Earlier work, such as SoRB (Eysenbach et al., 2019\) and related methods, has explored this direction.
>
> We agree and do not claim the invention of graph-based planning. Instead, our contribution lies in the techniques needed for successful graph-based planning at test time for offline GCRL. SoRB operates in the online RL setting with a growing replay buffer and trains a specialized distance function using distributional Q-learning with ensembles to obtain reliable distance estimates. It then applies a hard distance threshold to prune long edges. TTGS, by contrast, requires no gradient-based training and works with any frozen offline GCRL agent and its existing value function. This immediate applicability to a wide range of base learners (five in our experiments, with consistent gains across all of them) without requiring specialized training is, in our view, the key conceptual difference. TTGS also replaces SoRB's hard threshold with a soft penalty that preserves graph connectivity, which our ablations show is critical to combat noise (Figure 4a, Appendix E).
>
> > The comparison with GAS and CompDiffuser is asymmetric, as these methods require additional training. Including other test-time-only planning methods or explicitly noting this difference would improve the comparison.
>
> Thank you for raising this point. We acknowledge this difference in Appendix A and note that TTGS operates under strictly more restrictive constraints, requiring no additional training, no generative models, and no online interaction, yet achieves comparable performance. To the best of our knowledge, TTGS is the first method that operates purely offline and at test time in the GCRL setting, so no direct baseline in this category exists. We will make this distinction more prominent. Exploring alternative test-time-only planning paradigms (e.g., tree search with learned world models, RRT-based planning) under the same constraints is an exciting direction for future work.
>
> > Computational scaling is not empirically validated... practical performance on large offline datasets is not shown.
>
> We provide new scaling experiments (HIQL, 8 seeds):
>
> | M | Success Rate | Build Time (s) | Path Time (s) |
> | :---- | :---- | :---- | :---- |
> | **humanoidmaze-giant-stitch** | | | |
> | 125 | 10.4 ±10.0 | 7.4 ±2.0 | 0.06 |
> | 250 | 13.3 ±11.6 | 5.9 ±1.1 | 0.07 |
> | 500 | 42.4 ±22.9 | 5.0 ±1.5 | 0.14 |
> | 1000 | 60.0 ±19.6 | 6.9 ±1.7 | 0.24 |
> | 2000 | 72.8 ±13.0 | 10.7 ±1.7 | 0.44 |
> | 4000 | 76.7 ±18.1 | 26.2 ±1.5 | 0.84 |
> | 8000 | 78.7 ±12.8 | 99.8 ±8.7 | 1.31 |
> | **antmaze-giant-stitch** | | | |
> | 125 | 29.2 ±8.3 | 9.3 ±4.2 | 0.05 |
> | 250 | 29.9 ±12.4 | 8.2 ±4.4 | 0.06 |
> | 500 | 51.6 ±12.1 | 3.7 ±0.4 | 0.10 |
> | 1000 | 53.6 ±16.8 | 5.8 ±1.4 | 0.20 |
> | 2000 | 70.9 ±10.6 | 10.2 ±1.3 | 0.39 |
> | 4000 | 76.5 ±5.2 | 25.5 ±0.9 | 0.69 |
> | 8000 | 81.1 ±2.5 | 88.6 ±2.5 | 1.13 |
>
> Performance increases with the number of samples. Build time grows quadratically but remains under 2 minutes even at $M{=}8000$. Path computation stays under 1.5s.
>
> > Graph construction depends on access to the full offline dataset at test time.
>
> No, access to the full offline dataset is not required. We merely require access to a small subset of the original data (we use M=4000, which is 0.4% of the dataset). When the policy is deployed by the same user/organization that trained it, the data is immediately available. Otherwise, requesting a small sample or collecting one through limited interaction is feasible. We will discuss this more explicitly in the limitations.
>
> > When the method fails, what are the primary failure modes?
>
> We analyzed 100 failed episodes across 4 agents (HIQL, QRL, GCIQL, OTA) and 2 environments (antmaze-giant-stitch, humanoidmaze-giant-stitch). All navigation failures look like base policy failures: the agent gets stuck near a subgoal (e.g., humanoid falls) or passes near the goal without reaching it. We saw no paths through walls or coverage gaps. On manipulation, evaluation goals lie outside the data distribution, creating a manifold gap; TTGS defaults to the base policy (Appendix C).
>
> We hope we have addressed all raised concerns with new experiments (which we will include in the revised paper) and planned revisions. If any remain, we would be happy to discuss them further. We kindly ask the reviewer to consider updating their score in light of these responses.

---

> > ### Author Rebuttal · Reviewer_1VSa · 2026-04-03
> >
> > I thank the authors for a thorough rebuttal. Several of my concerns are now fully resolved. However, some concerns remain partially addressed:
> >
> > * Narrow evaluation scope vs. framing: The decision to move manipulation results into the main text and revise the abstract is a positive step. However, the claim that TTGS 'does not degrade performance' on manipulation tasks sets a minimal standard, as the method has negligible effect in this setting. The title and conclusion continue to imply broader applicability than is supported by the current experiments. The contribution would be clearer and more credible if the method were explicitly positioned as targeting long-horizon navigation in offline GCRL. Please clarify how the framing will be adjusted beyond the abstract.
> > * Conditions under which TTGS is effective: The qualitative guidance that TTGS is beneficial when the base policy is reliable at short range but fails at longer horizons is reasonable, but lacks precision. The method has minimal impact on manipulation tasks and substantial effect on navigation, indicating a sharp transition in effectiveness. Is there a quantitative metric that could predict in advance whether TTGS will provide gains on a new task?

---

> > > ### Author Response · Authors · 2026-04-07
> > >
> > > We thank Reviewer 1VSa for the thorough follow-up.
> > >
> > > > Narrow evaluation scope vs. framing...
> > >
> > > We agree and will narrow the framing throughout the paper. In the **abstract**, we will qualify that the primary gains are on long-horizon locomotion. In the **introduction**, we will state that TTGS is most impactful when (1) the task requires long-horizon stitching, (2) the base policy is locally reliable, and (3) the dataset covers intermediate states. We will note that these conditions hold for navigation but not for the tested manipulation tasks. In the **conclusion**, we will replace broad language with a precise summary: TTGS provides substantial navigation improvements and defaults to base policy behavior when its conditions are not met.
> > >
> > > > Conditions under which TTGS is effective... Is there a quantitative metric that could predict in advance whether TTGS will provide gains on a new task?
> > >
> > > We propose two complementary predictors and provide new experimental evidence for the first.
> > >
> > > **1. Dataset coverage: largest hop ratio (rollout-free).** TTGS decomposes a long-horizon task into short hops through dataset states. We propose a rollout-free diagnostic: after running Dijkstra on the TTGS graph, compute the *largest hop ratio*
> > >
> > > $$\frac{\max\bigl(d(s_0, p_0),\; d(p_0, p_1),\; \ldots,\; d(p_{L-1}, p_L),\; d(p_L, g)\bigr)}{d(s_0, g)}$$
> > >
> > > A low ratio means every hop is small relative to the total task, which is exactly the regime where TTGS helps. A high ratio means at least one hop spans a large fraction of the distance, indicating insufficient coverage.
> > >
> > > We computed this metric across all evaluation tasks using GCIQL (8 seeds), sorted by hop ratio:
> > >
> > > | Dataset | Hop Ratio | GCIQL | +TTGS | $\Delta$ (pp) |
> > > | --- | --- | --- | --- | --- |
> > > | pointmaze-giant-stitch | 0.041 | 0.0 | 98.0 | **+98.0** |
> > > | antmaze-giant-stitch | 0.080 | 0.0 | 32.7 | **+32.7** |
> > > | humanoidmaze-giant-stitch | 0.112 | 0.2 | 0.2 | 0.0 |
> > > | humanoidmaze-medium-stitch | 0.140 | 14.0 | 20.2 | +6.2 |
> > > | antmaze-medium-stitch | 0.207 | 29.5 | 53.0 | +23.5 |
> > > | pointmaze-medium-stitch | 0.217 | 18.2 | 44.0 | +25.8 |
> > > | scene-play | 0.458 | 50 | 52 | +2 |
> > > | cube-triple-play | 0.470 | 4 | 4 | 0 |
> > > | puzzle-4x6-play | 0.620 | 10 | 10 | 0 |
> > >
> > > Manipulation tasks hop ratios are 4-6 times higher than navigation tasks, confirming that the dataset lacks intermediate waypoints. Among locomotion tasks, giant mazes have the lowest ratios and the largest TTGS gains. This metric requires only the offline dataset and value function, without any rollouts.
> > >
> > > **2. Base policy local competence (requires rollouts).** Even with good coverage, TTGS needs the base policy to execute short hops reliably. Figure 2c shows navigation policies achieve high short-range success but low long-range, which is ideal for TTGS. We believe the case of humanoidmaze-giant-stitch (low hop ratio, zero improvement) to be an example of low base policy local competence: although coverage is good, the humanoid embodiment makes short-hop execution unreliable for GCIQL, unlike pointmaze and antmaze where the simpler embodiment allows reliable local control.
> > >
> > > Estimating local competence requires rollouts, at which point it might be simpler to evaluate TTGS directly. We therefore recommend the following procedure for a new task: (1) compute the hop ratio as a rollout-free preliminary check; if it exceeds ~0.4, TTGS is unlikely to provide gains; (2) if it is low, run TTGS directly, since the method defaults safely to base policy behavior when its conditions are not met.
> > >
> > > We will add both diagnostics and the table to the paper. We hope these concrete predictors and the planned framing revisions fully address the reviewer's remaining concerns.

---

### Official Review · Reviewer_Zg7J · 2026-03-10

**Soundness:** 3
**Presentation:** 2
**Significance:** 3
**Originality:** 2
**Overall Recommendation:** 4
**Confidence:** 5

**Summary:**

This paper proposes Test-Time Graph Search (TTGS), a planner for offline goal-conditioned reinforcement learning (GCRL). The key observation is that pretrained goal-conditioned value functions, while unreliable for long-horizon planning, encode locally consistent distance estimates that can support graph-based search. TTGS samples states from the offline dataset as graph vertices, assigns edge weights by inverting the learned value function into step-distance estimates, and runs Dijkstra's algorithm to produce a sequence of subgoals for frozen low-level policies. An adaptive subgoal selection scheme feeds the farthest reachable waypoint to the policy at each step. On OGBench locomotion tasks, TTGS dramatically improves success rates.

**Compliance With Llm Reviewing Policy:**

Affirmed.

**Final Justification:**

resolved most concerns through rebuttals, only remaining concern is the narrow scope of the paper compared to extensive related work in this area.

**Key Questions For Authors:**

In visual domains, does TTGS require ground-truth low-dimensional state for the goal-reaching reward? How does this affect generality?
How does TTGS perform with substantially weaker value functions where even local estimates are unreliable?

**Limitations:**

Yes

**Strengths And Weaknesses:**

Strengths:

- Simple, clearly explained method with only three hyperparameters and no retraining. Immediately practical.
- Impressive navigation results. Gains like HIQL going from 0% to 81% on pointmaze-giant-stitch, consistent across five base learners, demonstrate this is a general-purpose wrapper.
- Useful practical insight: complex auxiliary training (generative planners, distributional ensembles) may be unnecessary for many OGBench navigation tasks.
- Thorough ablations.

Weaknesses:

- The problem setting is narrow. TTGS assumes locally accurate value functions, good dataset coverage along solution paths, and a known goal-reaching metric. These sidestep the central GCRL challenges: sparse/unknown rewards, exploration for coverage, unknown metrics, and subgoal selection under non-uniform coverage. Substantial prior work on graph-based planning for goal-conditioned agents has tackled these harder problems — discovering graph structure and skills jointly, hierarchical goal-conditioned learning across abstraction levels, skill discovery without reward signals, and planning under poor coverage. While the offline setting is distinct, TTGS operates where the hardest parts are already solved. The contribution reduces to showing Dijkstra suffices in this regime — practically useful but conceptually limited.
- The formulation in Section 2 defines the MDP without rewards, then introduces ||s-g|| without specifying what the norm operates on. This is fine for low-dimensional state, but the paper evaluates on visual domains where pixel-space Euclidean distance is meaningless. The reward presumably uses privileged ground-truth state — this should be stated explicitly, as it affects generality claims.
- The related work should engage with the broader planning literature. TTGS is a classical planning algorithm (graph + shortest-path search) applied on top of RL. Classical planning, motion planning, and task-and-motion planning all deal with similar graph-based search and the paper should position itself relative to these.
- The manipulation results belong in the main paper. The method working well on navigation but not manipulation is a key finding about its scope, not an extraneous detail.
- Baseline methods are introduced only by acronym without describing what they do or why they were chosen.
- Minor: Several prominent terms are vague: "latent capabilities," "locally consistent geometric structure," "global errors." The central claim about value functions' local reliability deserves a precise definition.

---

> ### Author Rebuttal · Authors · 2026-03-31
>
> We thank Reviewer Zg7J for the detailed feedback. We address each concern below.
>
> > The problem setting is narrow. TTGS assumes locally accurate value functions, good dataset coverage along solution paths, and a known goal-reaching metric. These sidestep the central GCRL challenges: sparse/unknown rewards, exploration for coverage, unknown metrics, and subgoal selection under non-uniform coverage.
>
> We believe this characterization conflates the offline GCRL setting, which is considered in our work, with general GCRL. We address each listed challenge:
>
> - **Sparse/unknown rewards:** In offline GCRL, the dataset contains only trajectories of states and actions with no reward labels. All base learners we evaluate construct their training signal via hindsight relabeling, a self-supervised procedure that does not require any external reward. This is not an assumption specific to TTGS; it is the standard offline GCRL formulation shared by all prior work we compare against (Park et al., 2025; Wang et al., 2023; Zhou & Kao, 2025; Ahn et al., 2025).
> - **Exploration for coverage:** Exploration is not available by definition in the offline setting. The learner must work with whatever data is provided. We evaluate across three data regimes (navigate, stitch, explore) with varying coverage quality.
> - **Unknown metrics:** TTGS does not require a known metric. It derives distances from the learned value function, which is trained without access to any ground-truth metric.
>
> The absence of online interaction is a constraint that makes the problem harder, not easier.
>
> Regarding locally accurate value functions: we do not assume access to particularly good value functions. We take five different base learners as-is, with their published hyperparameters, and show consistent gains across all of them. The fact that TTGS improves agents as different as HIQL, QRL, GCIQL, SAW, and OTA suggests it does not depend on any specific quality of the value function beyond what these standard methods already provide. If locally accurate value functions were a strong assumption, we would expect TTGS to fail on at least some of these learners.
>
> That said, we agree the contribution is practically focused: given standard offline GCRL agents that already exist, TTGS provides a simple, training-free way to dramatically improve their long-horizon performance. We believe this is a valuable finding because it changes how practitioners should approach long-horizon offline GCRL.
>
> > The formulation in Section 2 defines the MDP without rewards, then introduces $||s-g||$ without specifying what the norm operates on... The reward presumably uses privileged ground-truth state.
>
> The norm $||s-g|| < \epsilon$ is the benchmark's evaluation criterion, not an input to TTGS. TTGS only needs a distance signal derived from $V(s,g)$ at test time and doesn't have access to privileged information. The base learners train without privileged information, too, using hindsight relabeling from the offline dataset. Since no online rollouts occur, no goal-reaching check is needed during training. We will clarify this.
>
> > The related work should engage with the broader planning literature.
>
> We agree and will position TTGS relative to classical planning, motion planning, and task-and-motion planning in the revision. TTGS shares the graph-search structure with these fields, but differs in a key aspect: edge costs are obtained from learned, noisy value functions rather than known dynamics or collision checkers. This motivates TTGS's soft penalty, which has no direct analogue in settings where feasibility can be verified exactly.
>
> > The manipulation results belong in the main paper. / Baseline methods are introduced only by acronym.
>
> We will move manipulation results to the main text and discuss both locomotion and manipulation results more thoroughly, including clarifying in the abstract that the primary gains are in locomotion. We have chosen HIQL, QRL, GCIQL because they were included in the OGBench evaluation, and SAW, and OTA for their strong performance. We will add brief descriptions of each baseline method, including full names, and clarify why we chose them.
>
> > Several prominent terms are vague: "latent capabilities," "locally consistent geometric structure," "global errors."
>
> We will tighten these definitions in the revision. By "locally consistent geometric structure" we mean that value-derived distances are approximately correct for nearby state pairs (within the trust region $\tau$), even when they are unreliable at longer horizons. We will make this precise.
>
> We hope we have addressed all the raised concerns. If any remain, we would be happy to discuss them further. We kindly ask the reviewer to consider updating their score in light of these responses.

---

> > ### Author Rebuttal · Reviewer_Zg7J · 2026-04-03
> >
> > - On rewards and metrics: I accept that the goal-reaching reward is standard in GCRL and shared across all baselines, so this is not a weakness specific to TTGS. However, I want to push back on the claim that GCRL involves "no reward labels." A reward function r = 1{‖s−g‖ < ε} absolutely exists in this pipeline, it's automatically computed rather than manually annotated, but it requires a defined distance metric over states and a threshold. If you didn't have something akin to reward functions or state-to-state distance metrics (which are a much stronger requirement than reward functions), then how did you get value functions? The assumptions enabling TTGS are stronger than the paper's framing suggests.
> >
> > - On planning literature: I appreciate the willingness to engage with classical planning, motion planning, and TAMP. Based on your framing, there are many RL papers that also do what you're saying, albeit for online settings. For e.g., "Goal-Space Planning with Subgoal Models" and "Intrinsically Motivated Discovery of Temporally Abstract Graph-Based Models of the World." The core structure of TTGS---graph construction from states, value-derived edge costs, shortest-path search, subgoal-conditioned policy execution---is well-precedented in this literature. The soft penalty and test-time-only application are useful practical refinements, but the paper should engage with these prior methods to clearly delineate what is new. This doesn't diminish the empirical contribution on OGBench, but the novelty should be characterized more precisely.

---

> > > ### Author Response · Authors · 2026-04-03
> > >
> > > We thank Reviewer Zg7J for the continued engagement.
> > >
> > > > On rewards and metrics: [...] A reward function r = 1{||s-g|| < epsilon} absolutely exists in this pipeline [...] it requires a defined distance metric over states and a threshold. [...] The assumptions enabling TTGS are stronger than the paper's framing suggests.
> > >
> > > We want to clarify an important distinction: $r = \mathbf{1}\{||s - g|| < \epsilon\}$ is **only the benchmark's evaluation criterion**. It is never accessed during training and never accessed by TTGS at inference time.
> > >
> > > All base learners train via **hindsight relabeling**, which requires no distance metric or threshold. We illustrate this with the per-step penalty convention used by HIQL, SAW, and GCIQL (the other base learners use different reward conventions but the same principle applies; see Appendix B for all mappings). For each transition $(s_t, a_t, s_{t+1})$, a goal $g$ is sampled from the dataset (the current state, a future trajectory state, or a random state). The reward is $r = 0$ if $g$ is the current state, and $r = -1$ otherwise, a pure index identity check with no norm involved. TD-learning then trains $V(s_{t}, g)$ to minimize the error against bootstrap targets $r_t + \gamma \cdot \mathbf{1}[s_t \neq g] \cdot V(s_{t+1}, g)$. Since the target at the goal is $0$, bootstrapping backward drives $V(s_t, g)$ toward $\approx -\sum_{i=0}^{k-1} \gamma^i$ for a state $k$ steps from $g$, a monotone function of step distance that emerges entirely from trajectory structure, without ever computing $||s - g||$. The only inputs are trajectory indices and the $-1$ per-step penalty or any other convention that derives training rewards from these indices.
> > >
> > > We agree the paper's framing was imprecise and will clearly separate the evaluation criterion ($||s - g|| < \epsilon$, privileged) from the training and inference pipeline (no privileged information) in the revision.
> > >
> > > > On planning literature: [...] "Goal-Space Planning with Subgoal Models" and "Intrinsically Motivated Discovery of Temporally Abstract Graph-Based Models of the World." The core structure of TTGS [...] is well-precedented in this literature. [...] the paper should engage with these prior methods to clearly delineate what is new.
> > >
> > > Thank you for highlighting these works, and we agree that the high-level structure of TTGS (graph construction from states, value-derived edge costs, shortest-path search, subgoal-conditioned policy execution) is well-precedented in this literature. We will expand the related work to discuss GSP and IM-DSG alongside SoRB (Eysenbach et al., 2019), which we already cite, and clearly position TTGS within this line of work.
> > >
> > > That said, these prior methods differ from TTGS in important ways. **GSP (Lo et al., 2024)** is a training-time method that learns four subgoal-conditioned models online and uses them for potential-based reward shaping to accelerate learning. **IM-DSG (Bagaria et al., 2025)** incrementally builds a skill graph through online exploration, learning option policies, novelty-driven expansion, and optimistic edge probabilities refined through repeated interaction. **SoRB** trains a specialized distributional distance function online and applies hard edge pruning. All three require online interaction and learn additional components.
> > >
> > > Our contribution is showing that this general idea can work **at test time, with no additional gradient-based training, on top of frozen offline GCRL agents**, and that this simple setup yields substantial performance improvements. Specifically, the novelty lies in: (1) the ability to operate purely at test time with zero components trained with gradient descent beyond the base agent's existing value function; (2) easy applicability on top of existing offline GCRL methods; (3) the soft penalty mechanism, which is necessary precisely because TTGS cannot verify edges via models or online execution (hard thresholds cause up to 99% graph disconnections, Appendix E); and (4) the empirical demonstration that standard offline GCRL value functions already encode sufficient local geometric structure for effective planning, making (1)-(3) viable without any specialized distance learning. We will characterize the novelty in these precise terms in the revision.
> > >
> > > To summarize our planned revisions: (1) clarify that $||s-g||<\epsilon$ is evaluation-only and explain hindsight relabeling precisely, (2) expand related work to discuss GSP, IM-DSG, SoRB, and broader planning literature, clearly delineating what is shared and what is new. We will incorporate all of these changes in the camera-ready revision. We kindly ask the reviewer to consider whether these clarifications address the remaining concerns.

---

### Official Review · Reviewer_K7zo · 2026-03-11

**Soundness:** 3
**Presentation:** 3
**Significance:** 3
**Originality:** 3
**Overall Recommendation:** 4
**Confidence:** 3

**Summary:**

Rather than designing complex auxiliary training mechanisms to overcome the noise in value functions, TTGS constructs a graph over offline dataset states, reinterprets existing goal-conditioned value functions as distance signals, and performs shortest-path search at test time. This dynamically provides a sequence of reachable intermediate subgoals to a frozen GCRL policy, substantially improving long-horizon goal-reaching capability without any retraining.

**Compliance With Llm Reviewing Policy:**

Affirmed.

**Final Justification:**

Thanks for the authors' responses, most of my questions have been resolved. However, due to the limited performance of TTGS on manipulation tasks, I will maintain my current positive score.

**Key Questions For Authors:**

## Questions

1. Value-derived distances are generally asymmetric and need not satisfy the triangle inequality. In the graph construction, do edge weights ever violate these properties in practice, and if so, how does this affect the validity of shortest-path planning?

2. The paper acknowledges that TTGS fails when intermediate states are absent from the offline dataset, particularly on manipulation tasks. Could the authors provide a more detailed discussion of potential solutions to this limitation, even at a theoretical level?

3. The results suggest non-trivial sensitivity to hyperparameters across environments. Could the authors provide any heuristics or guidelines for setting these parameters in a new environment？

**Limitations:**

Yes.

**Strengths And Weaknesses:**

## Strengths

**Soundness:** The authors reinterpret goal-conditioned values as distance signals, construct a graph over offline data, and combine soft penalties with adaptive subgoal selection for test-time shortest-path planning. Extensive experiments spanning 5 base learners, multiple maze scales, and diverse observation types (state-based and pixel-based) provide strong support for the method's generality. Ablation studies further confirm that both the soft penalty and adaptive subgoal selection are critical to the observed performance gains.

**Presentation:** The paper is clearly written overall, with an intuitive presentation of problem motivation, method decomposition, and experimental structure. The appendices are thorough, covering complete result tables, runtime analysis, hyperparameter settings, and value-to-distance mappings, providing a reasonable degree of reproducibility.

**Significance and Originality:** Prior graph search methods typically require online interaction or specialized distance function training. TTGS restricts the entire pipeline to the test phase, which constitutes a meaningful departure from existing paradigms. The paper treats existing goal-conditioned value functions as local geometric signals and, under a fully test-time, training-free setting, unlocks the latent potential of pretrained policies through soft-penalized graph search and adaptive subgoal selection.



## Weaknesses

1. The paper uses value functions as distances. However, value-derived distances generally do not satisfy the formal properties of a metric. They are not necessarily symmetric and may not obey the triangle inequality. This means the core edge weights in TTGS lack rigorous metric-theoretic guarantees, which is especially concerning given that value estimates tend to become noisier over longer horizons.

2. The method is highly sensitive to offline data coverage. TTGS retrieves and plans exclusively over dataset states, so whenever intermediate states between the start and goal are absent, no viable path exists in the graph. The paper observes this on manipulation tasks, where evaluation goals frequently fall outside the main training distribution. This is a fundamental methodological limitation, not merely an incidental experimental observation.

3. The paper introduces a soft penalty to address the problem of spurious shortcuts. However, using a hard threshold instead risks disconnecting the graph entirely. In other words, TTGS relies on an additional penalty function to balance two competing failure modes: over-trusting long edges versus fragmenting the graph. This reveals that the usability of the graph remains sensitive to errors in the distance estimates, which are largely driven by value function noise.

4. Experimental results indicate that TTGS exhibits notable sensitivity to its hyperparameters, which may limit the method's out-of-the-box applicability in new environments.

---

> ### Author Rebuttal · Authors · 2026-03-31
>
> We thank Reviewer K7zo for the thoughtful review and the recognition of TTGS's soundness, presentation, and originality.
>
> > Value-derived distances are generally asymmetric and need not satisfy the triangle inequality. In the graph construction, do edge weights ever violate these properties in practice, and if so, how does this affect the validity of shortest-path planning?
>
> This is a very good point, and we acknowledge that value-derived distances are not formal metrics (Section 3.1). To quantify this, we ran new experiments on triangle inequality violations across three base learners. We sample 1M triplets globally ("all") and separately 1M triplets where all pairwise distances fall under $\tau$ ("under $\tau$"), over 8 seeds:
>
> | Base Learner | Dataset | Violation Rate (all) | Violation Rate (under $\tau$) | Mean Positive Violation (all) | Mean Positive Violation (under $\tau$) |
> | :---- | :---- | :---- | :---- | :---- | :---- |
> | HIQL | humanoidmaze-giant-stitch | 0.2% | 13.4% | 585.7 | 3.1 |
> | HIQL | antmaze-giant-stitch | 1.3% | 9.1% | 587.8 | 1.2 |
> | QRL | humanoidmaze-giant-stitch | 0.7% | 0.3% | 0.001 | 0.002 |
> | QRL | antmaze-giant-stitch | 0.7% | 0.2% | 0.001 | 0.002 |
> | GCIQL | humanoidmaze-giant-stitch | 0.4% | 8.3% | 575.6 | 1.2 |
> | GCIQL | antmaze-giant-stitch | 5.6% | 8.2% | 574.9 | 1.2 |
>
> While violations do occur within the trust region, their magnitude is negligible: mean violations under $\tau$ are 0.002-3.1 steps, compared to 575-588 for unconstrained distances. This means that even when the triangle inequality is violated locally, the errors are small enough that they do not meaningfully distort shortest-path planning. This is consistent with TTGS's strong empirical performance across all base learners despite their different violation profiles. We will add this analysis to the paper.
>
> > TTGS fails when intermediate states are absent from the offline dataset, particularly on manipulation tasks. Could the authors provide a more detailed discussion of potential solutions to this limitation?
>
> We agree this is a fundamental constraint of any method planning over fixed data. We note two nuances: (1) sometimes intermediate states exist in the dataset, but the value function fails to recognize them as reachable due to long-horizon estimation errors. Improving value learning is complementary to TTGS. (2) When coverage gaps are detected (e.g., paths with too few intermediate waypoints), generative models trained on the offline data could synthesize additional states. For navigation, we performed a manual analysis of 100 failed episodes and found that the dominant failure mode is the base policy failing to reach a close subgoal, not missing graph coverage. We will expand the discussion of failures in the paper.
>
> > The results suggest non-trivial sensitivity to hyperparameters across environments. Could the authors provide any heuristics or guidelines for setting these parameters in a new environment?
>
> We achieved our strong performance without extensive hyperparameter sweeps. For most experiments, we tested 1-2 settings per $(\tau, T)$ pair; SAW used a single configuration across all tasks with no tuning. Our heuristic (detailed in Appendix H): set $\tau$ by visually inspecting the shortest paths on the graph. If paths are excessively fragmented into many short hops, $\tau$ is too small; if paths contain many unreliable long edges, $\tau$ is too large. Then set $T = 2\tau$. We will highlight this procedure in the main text.
>
> > TTGS relies on an additional penalty function to balance two competing failure modes... This reveals that the usability of the graph remains sensitive to errors in the distance estimates, which are largely driven by value function noise.
>
> The soft penalty is specifically designed for this: rather than removing noisy edges (which fragments the graph), it assigns them a high cost, preserving connectivity while steering the planner toward reliable short hops. The ablation in Figure 4a confirms that this outperforms both no-penalty and hard-threshold variants. Empirically, TTGS improves all five base learners we tested despite their substantially different value function characteristics, suggesting that the soft penalty is effective across a range of noise levels. We agree that improving value function quality remains a valuable complementary direction.
>
> We hope we have addressed all the raised concerns. If any remain, we would be happy to discuss them further. We kindly ask the reviewer to consider updating their score in light of these responses.

---

> > ### Author Rebuttal · Reviewer_K7zo · 2026-04-02
> >
> > Thanks for the authors' responses. My question has been resolved. However, due to the limited performance of TTGS on manipulation tasks, I will maintain my positive score.

---

### Decision · Program_Chairs · 2026-04-30

**Decision:**

Accept (regular)

**Comment:**

The paper introduces Test-Time Graph Search (TTGS), a training-free wrapper for offline goal-conditioned RL. TTGS constructs a graph over a subsample of dataset states, derives edge costs from a pretrained goal-conditioned value function using a soft penalty to handle long-range wormhole shortcuts, runs Dijkstra at test time, and feeds adaptive waypoints to a frozen base policy. On OGBench, TTGS delivers consistent and often dramatic improvements on long-horizon locomotion tasks (e.g., HIQL on pointmaze-giant-stitch: 0 → 80.9%; humanoidmaze-giant-stitch: 4.4 → 78.1%; antmaze-giant-stitch: 0 → 76.5%), and these gains replicate across five distinct base learners (HIQL, QRL, GCIQL, SAW, OTA).

Reviewers unanimously recognized the strong empirical results and the practical value of a training-free wrapper. The rebuttal added two pieces of evidence I consider important for the paper’s final value. First, a clean L2-kNN landmark baseline: planning over dataset states with purely geometric distances can work (up to 66% on antmaze-giant-stitch at k=20), but L2-kNN is brittle across k and environment – 3.2% at k=2 due to disconnection, 18.8% at k=20 due to wormholes, optimal k differs across tasks – whereas TTGS consistently reaches 76-77% through the soft penalty and full-graph construction. This experiment cleanly isolates that TTGS’s contribution is how to do graph search over value-derived distances, not that graph search helps. Second, a hop-ratio predictor (ratio of the largest subgoal hop to the start-goal distance) that quantitatively tells a practitioner in advance whether TTGS will help on a new task, with strong correlation to the actual gain (0.04 hop ratio → +98 pp on pointmaze-giant-stitch; 0.6 hop ratio → 0 pp on puzzle-4x6-play). This directly answers the regime-characterization question raised by reviewer 1VSa and partially by reviewer K7zo.

Two conceptual concerns remain and deserve explicit treatment:

Reviewer Zg7J correctly noted that the formulation in Section 2 introduces ||s-g|| < ε without being clear that this is a benchmark-side evaluation criterion rather than something consumed by the training pipeline or by TTGS. The authors’ follow-up walks through the hindsight-relabeling TD update (per-step -1 penalty if the relabeled goal is not the current state, bootstrapping backward toward 0 at the goal) and establishes that no norm is ever computed during training. This is technically correct and settles the privileged-information concern, but the manuscript must carry this clarification in the final version, not the discussion.

Reviewer Zg7J also asked for explicit engagement with Goal-Space Planning (Lo et al., 2024) and IM-DSG (Bagaria et al., 2025) alongside SoRB. The authors’ delineations are precise – GSP is a training-time reward-shaping method with four learned subgoal-conditioned models; IM-DSG builds a skill graph through online novelty-driven exploration; TTGS is unique in operating purely at test time, with no gradient steps beyond the base learner’s existing value function, on top of frozen offline GCRL agents. These distinctions must appear in the related-work section.
In addition to the mentioned works, authors should include [Bagatella et al NeurIPS 2023] https://openreview.net/forum?id=QlbZabgMdK as it is even closer to the current paper than the mentioned papers above.

Reviewer 1VSa correctly pushed the authors to narrow the framing beyond the abstract. The authors have committed to explicit language in the abstract, introduction, and conclusion specifying that TTGS helps when (1) the task requires long-horizon stitching, (2) the base policy is locally reliable, and (3) the dataset covers intermediate states – and that these conditions hold for navigation but not for the tested manipulation tasks. I expect this change in the camera-ready.

Reviewer K7zo’s triangle-inequality concern is cleanly defused: within the trust region τ, violation rates are 0.3-13.4% and violation magnitudes are 0.002-3.1 steps (vs ~575-588 at unconstrained scales), so the theoretical non-metricity does not materially corrupt shortest-path planning. This is a valuable addition.

I have read the full discussion, including the reviewer replies and follow-ups. My view is that the paper makes a clear, simple, and broadly useful contribution; the empirical evidence is strong, and the post-rebuttal additions (L2-kNN ablation, hop-ratio diagnostic, triangle-inequality quantification) materially strengthen it further. I recommend acceptance. The authors should ensure the camera-ready (i) moves manipulation results to the main text, (ii) narrows the framing throughout, (iii) incorporates the hop-ratio predictor and L2-kNN baseline, (iv) makes the evaluation-only nature of ||s-g|| < ε explicit in Section 2, and (v) the related work section discusses the relevant papers.